# Direct visualization of transcription-replication conflicts reveals post-replicative DNA:RNA hybrids

Henriette Stoy [1], Katharina Zwicky[1,5], Danina Kuster[1,5], Kevin S Lang[2,4], Jana Krietsch[1], Magdalena P. Crossley[3], Jonas A. Schmid [1], Karlene A. Cimprich [3], Houra Merrikh [2] & Massimo Lopes [1]✉

Transcription-replication collisions (TRCs) are crucial determinants of genome instability. R-loops were linked to head-on TRCs and proposed to obstruct replication fork progression. The underlying mechanisms, however, remained elusive due to the lack of direct visualization and of non-ambiguous research tools. Here, we ascertained the stability of estrogen-induced R-loops on the human genome, visualized them directly by electron microscopy (EM), and measured R-loop frequency and size at the single-molecule level. Combining EM and immuno-labeling on locus-specific head-on TRCs in bacteria, we observed the frequent accumulation of DNA:RNA hybrids behind replication forks. These post-replicative structures are linked to fork slowing and reversal across conflict regions and are distinct from physiological DNA:RNA hybrids at Okazaki fragments. Comet assays on nascent DNA revealed a marked delay in nascent DNA maturation in multiple conditions previously linked to R-loop accumulation. Altogether, our findings suggest that TRC-associated replication interference entails transactions that follow initial R-loop bypass by the replication fork.

In a healthy cell, transcription and replication are spatially and temporally separated across most organisms[1–4]; however, in pathological conditions, such as oncogene overexpression, this coordination is perturbed, leading to TRCs. Ultimately, TRCs contribute to genomic instability and drive tumorigenesis[5,6]. In bacteria, TRCs lead to increased mutagenesis and accelerated evolution[7,8]. TRCs can occur in two different orientations, depending on whether transcription and replication progress towards each other (head-on) or in the same direction (co-directionally), with head-on conflicts being far more deleterious than co-directional conflicts[9–12]. DNA:RNA hybrid formation and topological stress have emerged as major drivers of TRCs, especially in the head-on orientation, but the underlying molecular mechanisms remain elusive.

Head-on conflicts favor the formation of DNA:RNA hybrids, which can compromise genome integrity and are therefore counteracted by specialized helicases[13–20] and RNases[21–23]. DNA:RNA hybrids are generally thought to occur in the context of co-transcriptional R-loops, which are three-stranded nucleic acid structures, generated by the reinvasion of the nascent mRNA into the DNA duplex behind the RNA polymerase (RNAP). R-loops can stall RNAPs, creating a potential barrier to replication. In vitro studies have suggested that replisomes stall only temporarily at these R-loop-associated RNAPs. By evicting

[1]Institute of Molecular Cancer Research, University of Zurich, Zurich, Switzerland. [2]Vanderbilt University School of Medicine, Nashville, TN, USA. [3]Department of Chemical and Systems Biology, Stanford University School of Medicine, Stanford, CA, USA. [4]Present address: Department of Veterinary and Biomedical Sciences, University of Minnesota, Saint Paul, MN, USA. [5]These authors contributed equally: Katharina Zwicky, Danina Kuster. ✉e-mail: lopes@imcr.uzh.ch

the RNAP, the replisome can continue its progression[24,25]; this eviction step is facilitated by accessory helicases, such as Pif1 (ref. [26]). A recent study suggests that the DNA:RNA hybrid ahead of the fork could remain intact during helicase bypass, thereby being transferred behind the fork[25]. Accordingly, factors that assist fork restart and DNA-damage tolerance behind the fork also limit DNA:RNA hybrid accumulation[27,28]. Interestingly, both prokaryotic[24,25] and eukaryotic replisomes[26], although they travel on opposite strands of the fork[29], have the capacity to bypass an R-loop-based roadblock on the lagging strand. Together, these studies suggest that R-loop-associated RNAPs ahead of the fork might be only temporary impediments, and that TRC-induced replication interference extends to post-replicative processes. However, whether this is the case in vivo remains unknown.

Topological stress, in the form of negative and positive supercoiling, is a natural byproduct of both transcription and replication and is regulated by topoisomerases[30–32]. Interestingly, a recent study has highlighted the requirement of topological stress for fork collapse at TRCs[33]. Limiting topological stress by gyrase or Topo IV in *Bacillus subtilis*[34], or topoisomerase 1 in human cells[32,33], is critical for TRC resolution. Mechanistically, topological constraints can arise in three ways: first, head-on progression of transcription and replication may lead to a build-up of excessive positive supercoiling in between the approaching machinery, impairing DNA unwinding[34]. Second, negative supercoiling behind the RNAP may favor R-loop formation, thereby stalling RNAPs and creating a barrier to replication[35]. Finally, topological constraints behind the replication fork—which are well documented, albeit largely ignored in mechanistic models[31]—may disturb proper coordination of DNA polymerases, primases, and Okazaki-fragment processing, blocking DNA synthesis. The relative importance of these components of topological stress—ahead and behind replication forks—in R-loop formation and TRC-associated replication interference is currently unclear.

The main limitation in our understanding of hybrid-associated TRCs is the lack of unambiguous techniques for studying R-loop formation within the context of ongoing replication. First, most available techniques rely on either the DNA:RNA-hybrid-specific antibody S9.6 (ref. [36]) or mutant RNase H enzymes[37], both of which cannot distinguish between an R-loop and a DNA:RNA hybrid structure[38]. Second, our current understanding of TRC is largely based on population assays, which correlate R-loop and hybrid formation and signs of replication fork progression, but fail to pin both events to the same DNA locus simultaneously. Here, we adapted a well-established workflow for the visualization of replication intermediates (RI) by electron microscopy (EM)[39], used it to visualize TRC-associated DNA:RNA hybrids, and addressed how they affect replication fork progression and architecture. We provide the first direct visualization of R-loops at TRC regions on the genome of two highly diverse organisms, that is bacteria and humans, and report R-loop frequency and size unambiguously, on the basis of single-molecule observations. In both bacterial and mammalian cells, DNA:RNA hybrid accumulation is linked to evident signs of replication stress, such as delayed fork progression and fork reversal. Although we did not detect canonical R-loops accumulating ahead of these transcriptionally challenged forks, we did observe that DNA:RNA hybrids and persistent discontinuities in the nascent strand—clearly distinct from Okazaki fragments—accumulate on daughter duplexes, suggesting that novel molecular mechanisms are involved in transcription-dependent replication interference.

## Results

### Direct EM visualization of in vitro-generated R-loops

To establish an EM-directed workflow that allows R-loop visualization, we used an in vitro transcription system based on the pFC53 plasmid[40], which contains the R-loop-prone promoter region of the mouse *Airn* gene under the control of the T3 promoter (Extended Data Fig. 1a).

We confirmed the induction of transcription-dependent and ribonuclease H (RNase H)-sensitive R-loops by gel shift assay[41] (Extended Data Fig. 1b) and ensured that they remained stable during EM sample preparation (Extended Data Fig. 1c). Using native EM[39], we detected RNase H-sensitive R-loop structures on roughly 40% of circular and linear pFC53 fragments (Fig. 1a). These R-loop structures consist of a duplex DNA:RNA hybrid, indistinguishable from the surrounding double-stranded DNA (dsDNA), and a displaced single-stranded DNA (ssDNA) strand (Fig. 1b). R-loop size ranges from 40 base pairs (bp) to 670 bp (Fig. 1c). To determine whether R-loops localized to the *Airn* gene, we focused on the linearized fragment, where the R-loop is positioned asymmetrically. As shown in Figure 1d, R-loops are strongly clustered in the region that corresponds to the *Airn* gene.

In addition to visualization of the R-loop itself, we combined EM with S9.6-gold labeling of DNA:RNA hybrids (immuno EM). Binding of the S9.6-gold conjugate was detected on roughly 50% of circular and linear pFC53 molecules (Fig. 1e,f), with S9.6-gold binding clustering within the same region in which R-loops had accumulated in native EM analysis (Fig. 1g). S9.6-gold binding was both transcription-dependent and sensitive to RNase H treatment, confirming its specificity for co-transcriptional DNA:RNA hybrids (Fig. 1e). However, competition experiments—in which we gradually diluted the R-loop-containing pFC53 with a differently sized plasmid that lacks an R-loop—suggest that S9.6-gold specificity and selectivity drop with decreasing R-loop frequency (Extended Data Fig. 1d–f). As such, this immuno-EM approach is inherently limited and is applicable only to conditions of high R-loop enrichment (see below). Overall, we conclude that R-loops can be visualized in vitro by both native and immuno EM, while maintaining R-loop stability.

### R-loop visualization and quantification on human genomic DNA

We applied this EM approach to directly visualize R-loops within the human genome. We used the estrogen-dependent breast-cancer cell line MCF7, which has previously been shown to exhibit a strong enrichment of R-loops[42]. To induce R-loop formation, MCF7 cells were deprived of estrogen for 48 hours and then stimulated with 100 nM estradiol (E2) for 2 hours. This estrogen stimulation causes a strong transcriptional burst, which drives R-loop formation. To assess whether we could induce R-loop formation efficiently, we performed quantitative PCR with DNA–RNA immunoprecipitation (qPCR–DRIP) for selected loci, including genes that are transcriptionally activated by estrogen (*SLC7A5* and *GREB1*), as well as constitutively active genes that are prone to R-loop formation (*RPL13A*). In agreement with previous results, we consistently observed a strong increase in DRIP signal after E2 induction at both inducible loci (Fig. 2a and Extended Data Fig. 2a,b).

To visualize these R-loops by native EM, we extracted genomic DNA before and after the transcriptional burst. The extracted genomic DNA shows an RNaseH-sensitive S9.6 signal, which was increased after the addition of E2 (Fig. 2b,c and Extended Data Fig. 2c,d). Importantly, R-loops on extracted DNA remained stable across the multiple steps of EM sample preparation, including genomic DNA digestion (Extended Data Fig. 2e,f) and purification (Extended Data Fig. 2g,h). Using an automated high-throughput EM workflow (Extended Data Fig. 2i)[43], we screened 1,200–1,650 Mb of this genomic DNA for R-loop structures (Fig. 2d and Extended Data Fig. 3a). We found, on average, 0.19 and 0.26 R-loop structures/Mb genomic DNA before and after the transcriptional burst, respectively (corresponding with, on average, 614 and 830 R-loops/cell). The size of these R-loops ranged from 30 bp to 3,000 bp, with most R-loops being smaller than 200 bp and the smallest ones possibly escaping detection (Fig. 2e). Interestingly, although small (<300 bp) R-loops were not reproducibly induced by the transcriptional burst (Extended Data Fig. 3b), we observed a marked and consistent induction of longer R-loops (>300 bp) (Fig. 2f). The higher

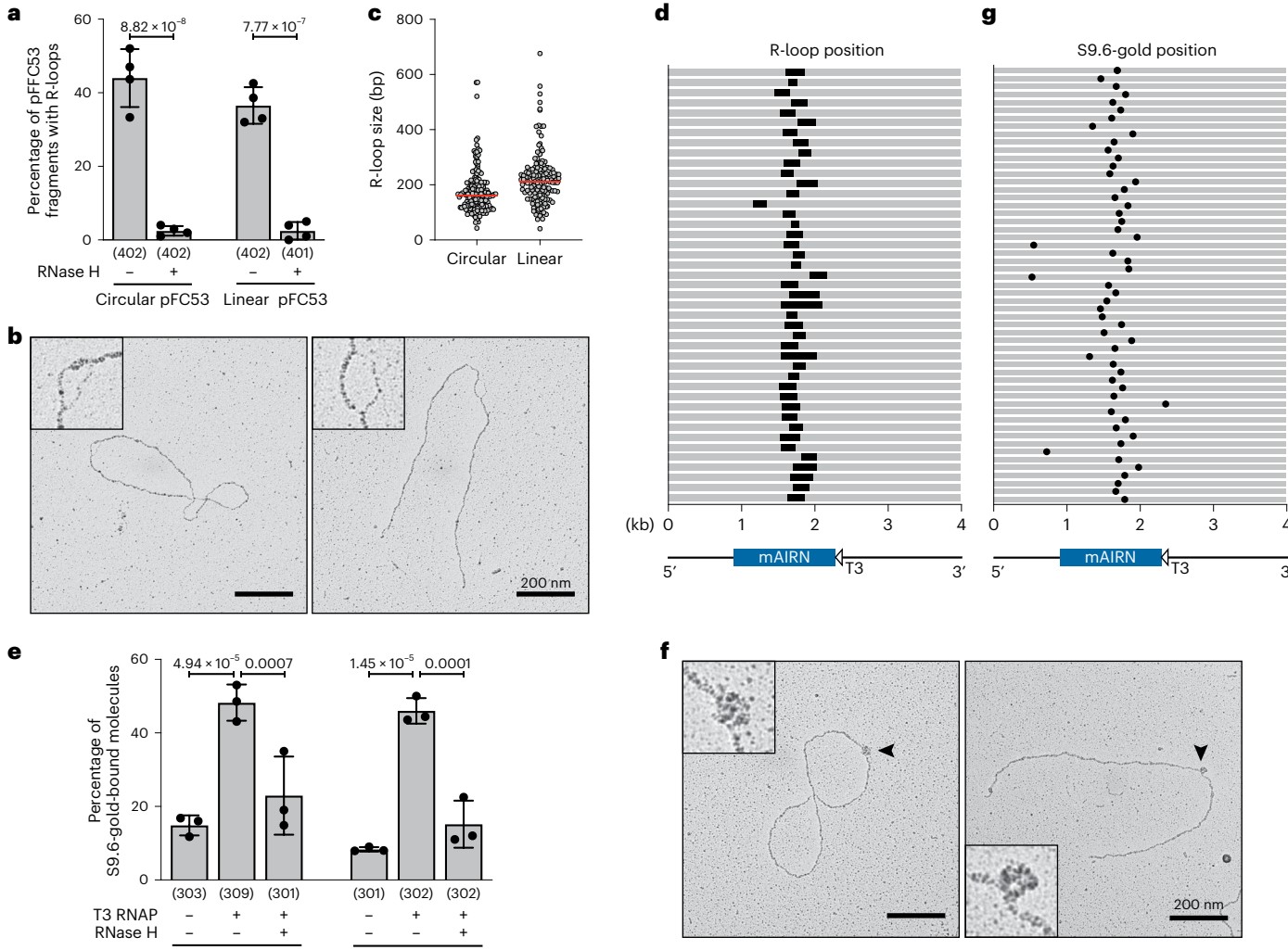

**Fig. 1 | Direct visualization of in vitro-generated R-loops using EM.**
**a**–**d**, Native EM analysis of in vitro-transcribed pFC53. Whenever indicated, samples were linearized and/or digested with 6 U/μg RNase H. At least 70 molecules were quantified per condition and replicate. **a**, R-loop frequency, shown as mean ± s.d., *n* = 4 independent biological replicates. Statistical significance was determined by ordinary one-way analysis of variance (ANOVA), followed by Sidak test. **b**, Representative electron micrographs. Scale bars, 200 nm. **c**, R-loop size; the median is indicated in red. **d**, R-loop position (black bar) on linearized pFC53 (gray). **e**–**g**, Immuno EM analysis of in vitro-transcribed

and S9.6-gold-labeled pFC53. Whenever indicated, samples were linearized and/or digested with 6 U/μg RNase H. At least 70 molecules quantified per condition and replicate. **e**, S9.6-gold binding frequency as mean ± s.d., *n* = 3 independent biological replicates. Statistical significance was determined by ordinary one-way ANOVA, followed by Sidak test. **f**, Representative electron micrographs. Scale bars, 200 nm. **g**, S9.6-gold binding position (black dot) on the linear pFC53 (gray). Numbers below *x*-axes in **a** and **e** indicate the total number of molecules analyzed in all replicates. In **d** and **g** the position of the R-loop forming *mAirn* gene is indicated in light blue below the graph.

levels of long R-loops in these conditions may also partially reflect increased stability or detectability of long hybrids in EM preparation and visualization.

Combining R-loop frequency and size information, we calculated the overall R-loop burden, that is the fraction of the genome that is involved in R-loop formation, assuming that R-loop formation was homogeneous within the cell population (Fig. 2g). R-loop burden reproducibly increased by twofold after the transcriptional burst, reaching a maximum of 0.009% of the human genome, meaning that about 290 kb of a cell's genome is involved in R-loop formation. Altogether, our EM approach provides the first direct visualization of R-loops in vivo and allows unambiguous quantification of R-loop frequency, size, and burden.

**Fork slowing and delayed nascent strand maturation at TRCs**
We next wondered whether and how R-loop formation affects replication fork progression. We used DNA fiber assays, in which ongoing DNA

synthesis is labeled sequentially by addition of nucleotide analogs, that is CldU and IdU, to follow fork speed before and after the transcriptional burst (Fig. 3a). We observed a decrease in CldU tract length specifically upon transcriptional burst induction, indicative of a replication fork slowdown. Overexpression of RNase H1 labeled with green fluorescent protein (GFP) (Extended Data Fig. 4a) restored normal fork progression upon the transcriptional burst (Fig. 3a and Extended Data Fig. 4b), linking the observed fork slowdown to accumulation of DNA:RNA hybrids. Moreover, downregulation of the DNA translocase ZRANB3 (Extended Data Fig. 4c)—previously implicated in the remodeling of replication forks into four-way junctions, that is reversed forks[44]—rescued fork speed upon E2-induced transcription (Fig. 3b and Extended Data Fig. 4d). A similar rescue of fork speed was observed upon PARP inhibition (Extended Data Fig. 4e), which has previously been shown to impair reversed fork stability[45], and was linked to a marked increase in γH2AX accumulation in stimulated cells (Extended Data Fig. 4f,g). Altogether, these data suggest that the transcriptional

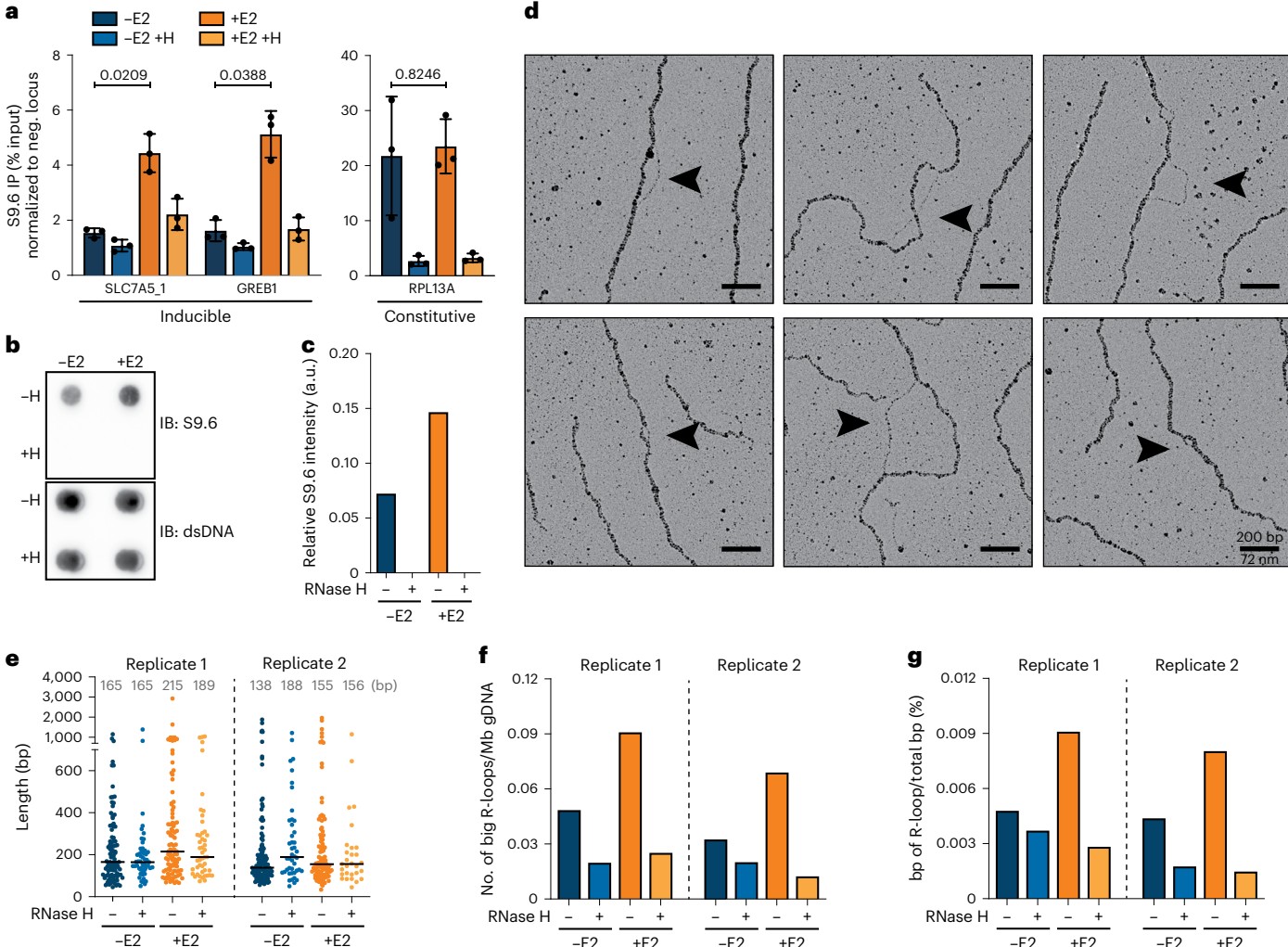

**Fig. 2 | EM-based visualization and quantification of R-loops on human genomic DNA upon estrogen-dependent transcriptional burst. a**, qPCR–DRIP analysis of MCF7 cells with or without 2 hours of E2 stimulation and with or without in vitro RNase H digestion (H) for representative E2-inducible and constitutive genomic loci. Data were normalized to a negative (neg.) locus and is shown as mean ± s.d., $n = 3$ independent biological replicates. Statistical significance was determined by unpaired two-tailed $t$-test. Inducible and constitutive loci differ significantly, as determined by two-way ANOVA ($P = 5.78 \times 10^{-7}$). **b**, Dot blot analysis of genomic DNA extracted from MCF7 cells with or without 2 hours of E2 stimulation and with or without in vitro RNase H digestion. Genome-wide hybrid accumulation was detected by S9.6 immunostaining (loading control: dsDNA). **c**, Quantification of integrated intensities in **b**; S9.6

signal was normalized to the respective dsDNA loading control. a.u., arbitrary units. **d**–**g**, EM analysis of R-loops on genomic DNA extracted from MCF7 cells with or without 2 hours of E2 stimulation and with or without in vitro RNase H digestion. This analysis was performed in two independent biological replicates (additional data in Extended Data Fig. 3). **d**, Representative electron micrographs of R-loops (indicated by the black arrows) found on gDNA from E2 stimulated cells. Scale bars, 200 bp/72 nm. **e**, Sizes of single R-loops in bp. Black lines and gray numbers indicate the median R-loop size. **f**, Frequency of R-loops that are >300 bp in size. Absolute numbers of R-loops were normalized to the total DNA content within the analyzed area. **g**, Total R-loop burden, calculated as the genomic fraction of bp involved in R-loop formation.

burst induces DNA:RNA hybrid- and fork-reversal-dependent replication fork slowing, ultimately limiting genomic instability. Of note, both RNase H1 overexpression and ZRANB3 depletion detectably accelerate replication fork progression even in the absence of E2 (Fig. 3a,b and Extended Data Fig. 4b,d), suggesting that MCF7 cells also experience a mild level of replication stress upon prolonged E2 depletion.

To assess whether this transcription-dependent replication interference is associated with any particular structural features, we conducted a thorough EM analysis of replication intermediates (RIs) after a transcriptional burst. We observed numerous small (<30 nt) ssDNA gaps that accumulated behind replication forks, at a close distance (<3 Kb) (Fig. 3c and Extended Data Fig. 4h,i, blue dots), possibly reflecting short discontinuities that are physiologically associated with lagging strand synthesis and maturation. A fraction of the observed

ssDNA gaps was unexpectedly large (>30 nt) and/or persisted at a notable distance from the fork regardless of E2 (Extended Data Fig. 4h,i, red dots), further suggesting that there is a form of endogenous replication stress in MCF7 cells. However, when we subjected this genomic DNA to in vitro RNase H treatment, we observed a marked increase in larger and/or persistent ssDNA gaps, which was specific to E2 addition (Fig. 3d,e). This effect was observed in both independent EM replicates (Fig. 3e and Extended Data Fig. 3h,i) and suggested that—despite endogenous RNase H activity—RNA molecules remain embedded in the duplex DNA up to 15 kb behind the fork. In vitro RNase H treatment degrades this embedded RNA, generating larger post-replicative ssDNA gaps (Fig. 3d,e).

Because of restriction digestion of genomic DNA, the analysis of nascent strand discontinuities by EM is limited to a few kb behind

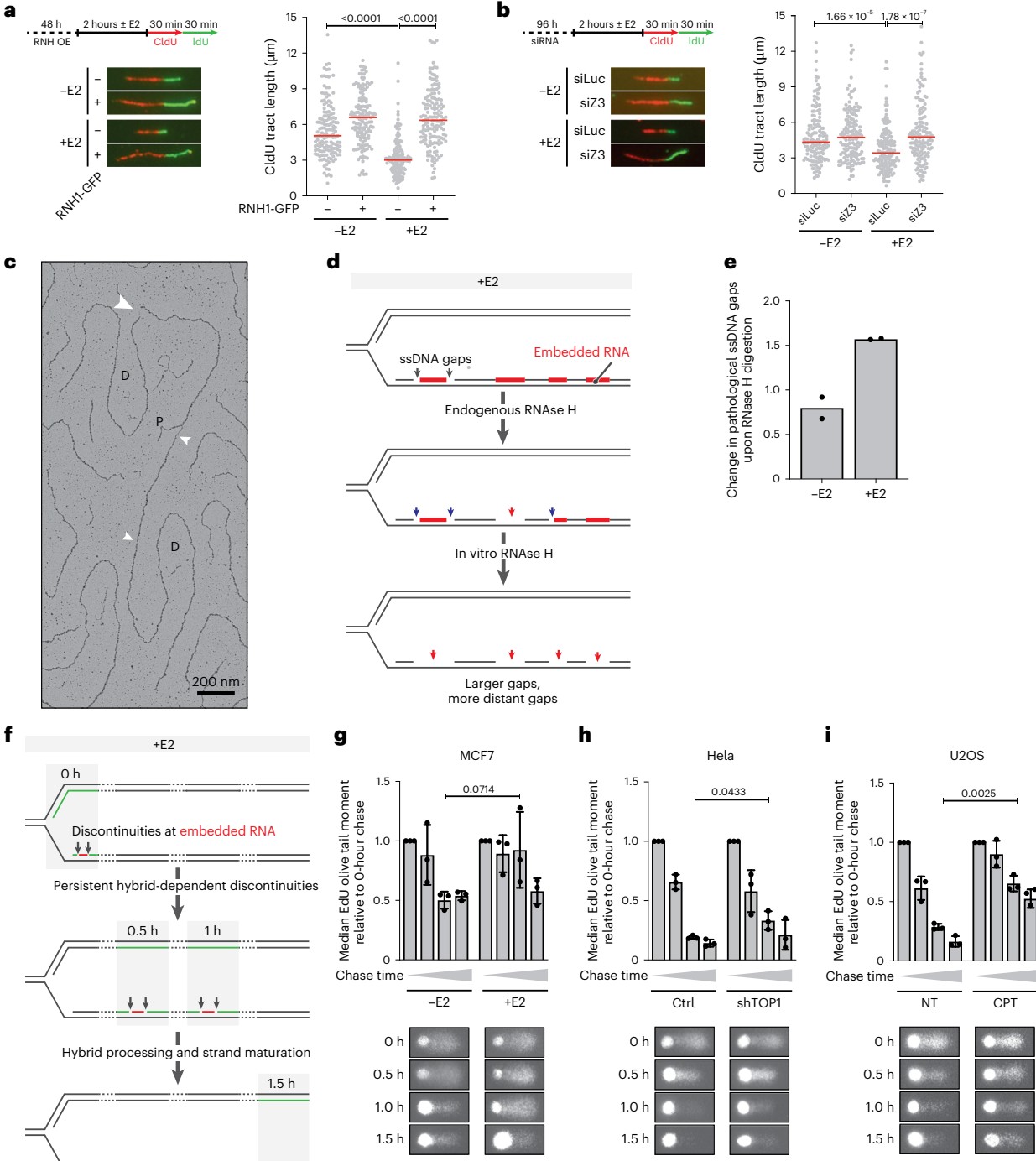

**Fig. 3 | Estrogen-dependent transcriptional burst results in replication stress and is associated with hybrid accumulation behind the replication fork.** **a,b**, DNA fiber assay of MCF7 cells with or without 2 hours of E2 stimulation, combined with 48 hours of transient RNH1-GFP expression (**a**) or 96 hours of short interfering RNA (siRNA, siLuciferase (siLuc) and siZRANB3 (siZ3)) transfection (**b**) prior to E2 treatment. Left, assay set-up (top), with representative DNA fibers (bottom). Right, quantification of CldU tract lengths (µm); at least 100 individual molecules were quantified per condition. Median fiber length is indicated in red. Statistical significance was determined by Kruskal–Wallis test, followed by Dunn's test. Triplicate data of these experiments are provided in Extended Data Figure 4b,d. **c–e**, EM analysis of pathological ssDNA gap formation on replicating genomic DNA from MCF7 cells with or without 2 hours of E2 and with or without in vitro RNase H digestion; 100 replication intermediates were quantified per condition (see Extended Data Fig. 4h,i). **c**, Representative replication fork with ssDNA gaps. P: parental; D: daughter; white arrow: ssDNA gap. Scale bar, 200 nm. **d**, Graphical model of changes in ssDNA accumulation,

induced by endogenous RNase H and/or in vitro treatment with recombinant RNase H. **e**, Relative change in pathological ssDNA gaps upon in vitro treatment with RNase H (based on red numbers in Extended Data Fig. 4h,i). **f–i**, EdU alkaline comet assay to identify discontinuities in nascent DNA strands. A detailed explanation of the assay set-up is provided in Extended Data Figure 4j. **i**, Graphical model of how the EdU alkaline comet assay can reveal persistent DNA:RNA hybrid-induced discontinuities. **g–i**, EdU alkaline comet assay in different cellular systems of transcription-replication conflicts. Top, median EdU olive tail moments—indicating amount and distance of EdU-labelled DNA migrating from the head region—normalized to the 0-hour chase time point. Bar graph shows mean ± s.d., n = 3 independent biological replicates; at least 30 single EdU-positive cells were analyzed per condition and replicate. Bottom, representative EdU comets. Statistical significance was determined by one-tailed t-test with Welch's correlation. **g**, MCF7 cells with or without 8 hours of E2 stimulation. **h**, Hela control and shTOP1 cells with 72 hours of 2 µg/ml doxycycline treatment. **i**, U2OS cells with or without 1 hour of 100 nM CPT treatment.

replication forks. Moreover, particularly small ssDNA gaps may escape detection. To overcome these limitations and to test the hypothesis that embedded DNA:RNA hybrids may lead to persistent discontinuities in nascent DNA, we optimized a recently published alkaline comet assay[46] to monitor nascent strand maturation, after pulse-labeling with EdU and a chase period of variable duration (0–1.5 hours; Extended Data Fig. 4j). In unperturbed conditions, our assay reveals physiological discontinuities during nascent strand synthesis, which are rapidly resolved upon nascent strand maturation (0.5–1.0 hour; Extended Data Fig. 4j). We reckoned that this assay may be used to reveal additional or persistent discontinuities, possibly associated with the accumulation of RNA molecules within nascent DNA (Fig. 3f). In agreement with previous assays, some nascent strand discontinuities persisted up to 0.5 hours in MCF7 cells, even in the absence of E2, likely marking endogenous replication stress (Fig. 3g). However, E2 addition drastically increased the half-life of nascent strand discontinuities, which were still detected 1 hour after EdU chase (Fig. 3g and Extended Data Fig. 4k). We then applied the same assays to other conditions that have previously been linked to DNA:RNA hybrid-dependent fork slowing, such as TOP1 depletion[33] or TOP1 inhibition by camptothecin (CPT)[47]. Strikingly, in all tested conditions, the persistence of nascent strand discontinuities was markedly increased when compared with that in the matched control conditions (Fig. 3h,i and Extended Data Fig. 4l–m), with discontinuities clearly visible in CPT-treated cells even 1.5 hours after EdU removal (Fig. 3i). Altogether, our data suggest that post-replicative DNA:RNA hybrids and persistent nascent strand discontinuities associate with delayed fork progression upon a transcriptional burst and other conditions of DNA:RNA hybrid accumulation.

## Fork slowing and reversal at locus-specific TRCs in bacteria

Our observations in MCF7 cells prompted us to explore a potential role for post-replicative DNA:RNA hybrids in replication interference at defined transcription-replication conflicts. To address this question, we took advantage of locus-specific head-on TRCs in the chromosome of the bacterial model system *Bacillus subtilis*. This *B. subtilis* strain carries an IPTG-inducible transcription unit that is oriented head-on to the bacterial origin (Fig. 4a). Previous work showed that expression of this engineered conflict increases hybrid formation and replication fork stalling within the same conflict region, in the absence of activity of RNase H III (Δ*rnhc* mutant strain)[9,48]. Combining this locus-specific TRC model system with our EM approach, we explored how replication forks progress through this TRC.

To isolate the conflict region from the rest of the bacterial genomic DNA, we designed a digestion/gel-extraction-based protocol (Fig. 4a) and obtained RIs of amenable size to perform EM analysis[49]. We first confirmed by qPCR–DRIP that DNA:RNA hybrids accumulate in the conflict region and remain stable throughout the digestion/gel-extraction procedure. In agreement with published results, we detected a significant and RNase H-sensitive increase in hybrid formation in the Δ*rnhc* strain (Fig. 4b). To assess replication fork progression through the conflict region, we analyzed the replicating conflict fraction of both wild-type (WT) and Δ*rnhc* strains, with or without IPTG treatment, by EM in three independent biological replicates (Fig. 4c,d and Extended Data Fig. 5a, additional replicates in Extended Data Fig. 5b–g). We collected all detectable RIs and sorted them by fragment length (Fig. 4c). In the WT-IPTG strain, a condition in which forks are not expected to slow down at the conflict locus, we observed RIs of all sizes between 5 kb and 15 kb. The lack of enrichment for the conflict fragment in this control condition impairs the identification and reliable EM analysis of the conflict region. Conversely, the WT strain with IPTG, as well as Δ*rnhc* with or without IPTG, showed a clear enrichment of RI with sizes matching the fragment of interest (marked in dark grey in Fig. 4c). This suggests that replication forks stall frequently in this conflict region when conflict is induced. The enrichment of conflict RIs in

Δ*rnhc* without IPTG is likely due to leaky transcription (which has been previously documented for the P$_{spank(hy)}$ promoter), in combination with perturbed clearance of hybrid accumulation. To assess where fork stalling occurs within this conflict region, we sorted the selected RIs and aligned them on the basis of the length of the daughter strand (Fig. 4d). In this analysis, replication fork stalling at a defined location should result in an accumulation of intermediates with a defined ratio of daughter (green) versus parental (gray) strand. In all three samples, however, we observed a uniform distribution of RIs across the conflict region. This suggests that transcription-replication interference does not block replication forks at a specific location, but rather leads to an overall slowing of replication fork progression throughout the conflict region, in line with published two-dimensional gel analysis[9]. Finally, we assessed replication fork remodeling. Conflict-induced RIs displaying reversed forks are shown in pink in Fig. 4d and sorted according to the position of the original junction (prior to reversal). We observed that hybrid accumulation in the Δ*rnhc* sample treated with IPTG was associated with a significant increase in replication fork reversal (Fig. 4e). Altogether, our data suggest that TRCs lead to replication fork stalling and reversal across the conflict region.

## Post-replicative DNA:RNA hybrids at head-on bacterial TRCs

Inspired by our evidence from the mammalian system (Fig. 3), we next investigated whether post-replicative DNA:RNA hybrids are present at TRC regions across species. We found that RIs from the WT bacterial strain display reproducibly higher levels of ssDNA gaps behind the fork than the Δ*rnhc* mutant strain, which lacks endogenous RNase H III activity (Fig. 5a).

To test whether RNA remains embedded in the conflict DNA in the Δ*rnhc* mutant-strain, we employed our S9.6-gold-based immuno EM approach (Fig. 1f)[49]. We compared S9.6-gold binding to the replicating conflict fraction with that to the linear conflict fraction. As a negative control, we included a bulk DNA fraction, which was extracted from a different region of the gel and is devoid of the transcriptional conflict; this fraction contained both linear and replicating DNA. As an internal control for S9.6-gold specificity, we spiked each extracted *B. subtilis* chromosomal fraction with an excess of R-loop-carrying, linearized pFC53 prior to S9.6-gold labeling and EM spreading. As depicted in Figure 5b (additional replicates are in Extended Data Figure 6a,b), *B. subtilis* DNA and pFC53 were identified by their respective sizes. S9.6-gold binding to pFC53 was constant between samples and replicates (Fig. 5c) and strongly reduced upon RNase H digestion, confirming that S9.6-gold labeling conditions were specific. In *B. subtilis*, S9.6-gold binding was observed on 30–40% of linear conflict intermediates, a reproducible increase compared with the bulk fraction. RNase H digestion markedly reduced S9.6-gold binding to the linear conflict intermediates and led to corresponding accumulation of ssDNA gaps (Fig. 5c–d and Extended Data Fig. 6c), suggesting that S9.6-gold does indeed label embedded DNA:RNA hybrids. Excitingly, S9.6-gold binding to *B. subtilis* RIs was significantly increased compared with both the linear conflict and bulk DNA, with 40–60% of RIs carrying at least one label (representative images, Fig. 5e). Importantly, S9.6-gold binding to bulk RIs is close to background levels and is not decreased upon RNase H treatment, excluding the concept that S9.6-gold binding reflects short DNA:RNA hybrids that are physiologically present at all RIs (for example, Okazaki fragments; Fig. 5c). S9.6-gold labels were found scattered throughout the conflict region (Fig. 5f and Extended Data Fig. 6d,e) on both parental and daughter strands. S9.6-gold binding to the parental DNA strand was detected on roughly 20% of the RIs. Because our native EM analysis (Fig. 4) did not reveal any detectable R-loop structures on parental DNA—even though DNA:RNA hybrids remain stable throughout the gel extraction procedure (Extended Data Fig. 7a,b)—we propose that S9.6-gold labeling ahead of forks reflects R-loop structures that are processed by removal of the displaced ssDNA and thus are indistinguishable from duplex DNA

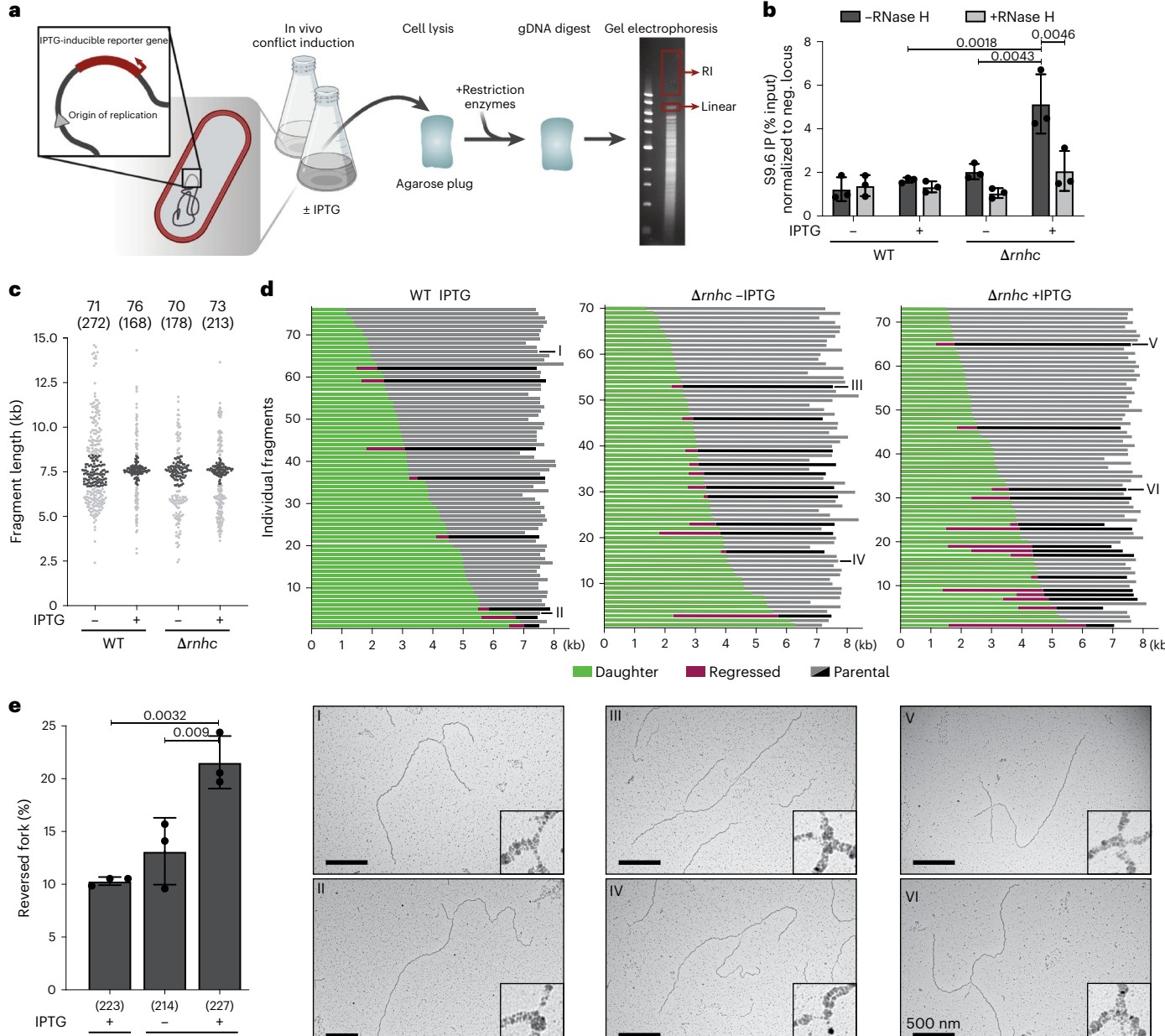

**Fig. 4 | Replication forks stall and reverse while facing a TRC in *B. subtilis*.**
**a**, Model system and experimental workflow; red boxes indicate the area from which linear and replicating conflict (RI) regions were extracted for further analysis. gDNA, genomic DNA. **b**, qPCR–DRIP analysis of accumulation of DNA:RNA hybrids within the conflict region in WT and *Δrnhc* mutant strains with or without IPTG induction and with or without in vitro RNase H digestion. Data were normalized to a negative locus and are shown as mean ± s.d., *n* = 3 independent biological replicates. Significance was determined by ordinary one-way ANOVA, followed by Sidak test. **c**–**e**, Native EM analysis of RIs extracted from the RI region, marked in Extended Data Figure 5a (data from additional replicates of this experiment are provided in Extended Data Figure 5b–g). **c**, Fragment lengths of all imaged RIs from one replicate. The two numbers

on top indicate the number of RI within the expected size range (top number, dark dots) and the number of total RIs imaged (in parentheses, all dots). **d**, Top, alignment of selected RIs, according to daughter strand length. Daughter strands are indicated in green, and parental strands in gray. Reversed forks are labeled in pink and black, with pink marking length of the regressed arm and black the length of the parental strand prior to reversal. Bottom, representative electron micrographs of normal (I, II, IV) and reversed (III, V, VI) forks. Scale bars, 500 nm. **e**, Fork reversal frequency, shown as mean ± s.d., *n* = 3 independent biological replicates (the corresponding fragment maps for the additional replicates can be found in Extended Data Fig. 5e,g). Numbers below the graph indicate the total number of molecules analyzed in all three experiments. Significance was determined by ordinary one-way ANOVA, followed by Sidak test.

in native EM. However, the majority of S9.6-gold labels on RIs bound to the daughter strands (Fig. 5f and Extended Data Fig. 6d,e), suggesting that there was extensive DNA:RNA hybrid formation on nascent DNA behind the replication fork. Altogether, these results strongly suggest that, in the absence of endogenous RNase H, RNA remains embedded in duplex DNA at transcription–replication conflicts.

## Discussion

Using a direct EM-based visualization approach, we measured frequency and size of R-loops on genomic DNA upon a transcriptional burst. Moreover, we directly monitored DNA:RNA hybrids at replication forks progressing 'head-on' through transcriptionally active regions, where we found them to markedly accumulate as 'embedded' hybrids

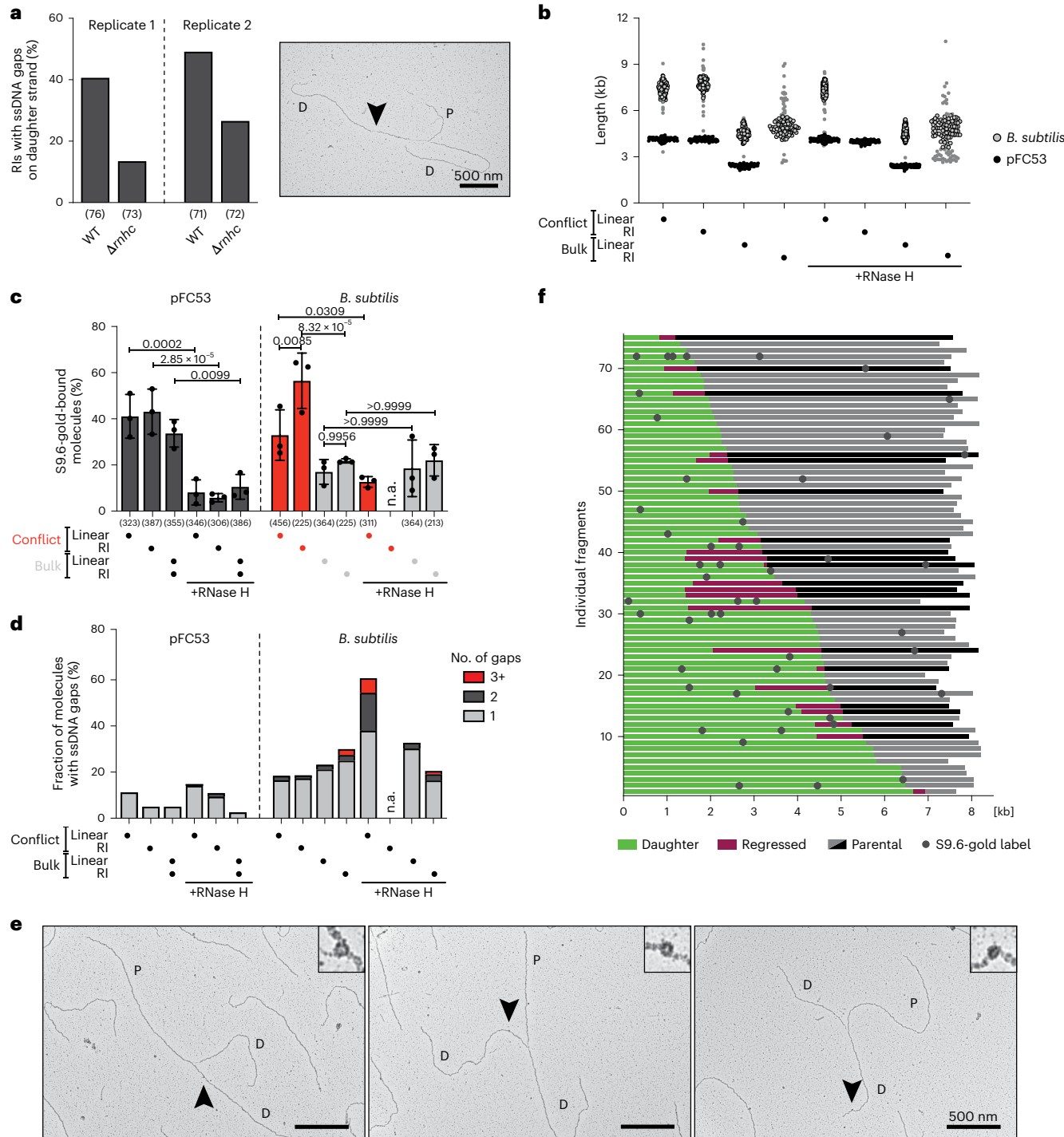

**Fig. 5 | DNA:RNA hybrids accumulate within replicating conflict DNA in _Bacillus subtilis_. a**, EM analysis of ssDNA gaps within replicating conflict DNA from IPTG-induced WT and Δ*rnhc* mutant strains. Left, fraction of RIs with at least one gap, from two independent biological replicates. The numbers below the graph indicate the number of molecules analyzed. Right, representative electron micrograph of a replication fork with a ssDNA gap. P: parental; D: daughter; R: regressed; black arrow: ssDNA gap. Scale bar, 500 nm. **b**–**f**, Immuno EM analysis of *B. subtilis* material, with pFC53 added as an internal control for S9.6-gold specificity. When indicated, samples were digested with RNase H in vitro. Of note, bulk RI and linear molecules were collected from the same sample. The RI fraction of *B. subtilis* was repeatedly lost during the RNase H digestion and had to be excluded from subsequent analysis (not applicable, n.a.). **b**, Length distribution of the imaged fragments. Black, pFC53; gray, *B. subtilis* material. At least 70 molecules were imaged for each fraction and fragment. Of note, the

relative frequencies of pFC53 and *B. subtilis* material displayed do not represent the frequency observed in the sample: pFC53 molecules were added in large excess. **c**, S9.6-gold-binding frequency of pFC53 and *B. subtilis* with or without RNase H, shown as mean ± s.d., *n* = 3 independent biological replicates. Numbers below the graph indicate the total number of analyzed molecules in all three experiments. Statistical significance determined by ordinary one-way ANOVA, followed by Sidak test. **d**, Frequency and numbers of ssDNA gaps detected in the analyzed molecules in **b**. **e**, Representative electron micrographs of S9.6-gold-labeled conflict RIs in *B. subtilis*. P: parental; D: daughter; R: regressed. Scale bars, 500 nm. **f**, S9.6-gold-binding position within the replicating conflict of *B. subtilis*. Green: daughter strand; gray: parental strand; black: parental strand of reversed forks prior to reversal; pink: regressed arm; dark gray dots: S9.6-gold label. Comparable maps of additional replicates are available in Extended Data Figure 6.

on newly replicated DNA. This observation prompts us to propose a new, integrated model of head-on transcription-replication interference (Fig. 6). We propose that R-loops forming behind an RNAP that approaches the replisome head-on are either processed into DNA:RNA hybrids and/or efficiently bypassed by the replisome. The DNA:RNA hybrid is thereby transferred to the lagging strand behind the replisome, and is typically removed and processed quickly, to allow continuous DNA synthesis. In case of excessive accumulation or impaired removal of these DNA:RNA hybrids behind the fork, replication fork progression is impaired and is frequently associated with delayed nascent strand maturation and fork reversal. Accordingly, post-replicative processes—such as nascent DNA resection—are prevented in the absence of RNase H2 in both yeast and human cells (P. Pasero, personal communication), further suggesting that DNA:RNA hybrids may act behind the fork to interfere with DNA replication. Although excessive transcription and defective removal of transient DNA:RNA hybrids led to similar molecular phenotypes across species in our EM analyses, our data do not exclude that other specific defects in RNA metabolism may alter this sequence of events in drastically different ways, possibly leading to accumulation of different pathological intermediates.

### R-loops form on human genomic DNA and can be visualized by EM

Using our optimized native EM approach[43], we provide the first direct visualization of R-loops on genomic DNA and quantify their frequency and size unambiguously at the single-molecule level. In unchallenged MCF7 cells, we detected an average of 614 R-loops/cell, with a size range of 30–3,000 bp. Previous sequencing-based studies estimated R-loop frequency to be around 300 R-loops/cell in unperturbed HeLa cells[50] and R-loop size to range between 100 and 450 bp, with some R-loops reaching up to 2.7 kb in size[51]. Our S9.6-independent approach provides visual support for these previous estimations and allows simultaneous retrieval of information on R-loop frequency and size, providing mechanistic clues about R-loop-mediated genomic instability. Upon an estrogen-induced transcriptional burst, for example, we observed that long R-loops (>300 bp) are most reproducibly increased, suggesting that defined subsets of R-loops are differently affected by specific perturbations and may differentially impact DNA replication and genome stability.

### DNA:RNA hybrids accumulation ahead of forks facing TRCs

Although our EM technique proved compatible with R-loop stability and could detect R-loops of sizes and frequencies comparable to those in previous studies, we did not detect an accumulation of R-loops ahead of replication forks, even at a locus-specific head-on TRC. However, S9.6-gold-dependent immuno EM of the conflict locus did reveal the presence of DNA:RNA hybrid structures ahead of forks, which are indistinguishable from duplex DNA in native EM. Our data show that, despite justified concerns about the use of S9.6 as an unambiguous readout of DNA:RNA hybrids[38], rigorous internal controls for antibody specificity allow it to be used as a reliable, potentially revealing research tool. We propose that these DNA:RNA structures arise from R-loop processing by topoisomerases and/or structure-specific nucleases[18,32]. Torsional stress accumulating at sites of transcription-replication interference is reportedly addressed by topoisomerases[32–34]. By cleaving the displaced strand of the R-loop, topoisomerases/nucleases may locally relieve torsional constraints and convert the three-stranded R-loop into a DNA:RNA hybrid structure. Although R-loop incision may expose cells to the risk of fork collapse and chromosomal breakage[18,33], it has recently been shown that cleavage-religation cycles are actively contributing to fork progression and restart at TRCs[47]. This suggests that nucleolytic processing may in fact be physiologically linked to R-loop bypass. We cannot exclude, however, that a fraction of R-loops escapes processing, yet remains undetected in our EM analysis. In line with in vitro observations[25], these R-loops may be bypassed very efficiently

by the replisome, preventing detectable accumulation of R-loops and forks in the same restriction fragment. Whether through processing or bypass, we consider it likely that an R-loop ahead of the fork can be efficiently transferred behind the fork as a DNA:RNA hybrid, without markedly affecting the fork progression rate (Fig. 6).

### Origin and clearance of post-replicative DNA:RNA hybrids

At a bacterial locus-specific TRC impairing fork progression, S9.6-gold binding was most frequently observed behind the replication fork. Accordingly, in all tested genetic conditions that have previously been associated with DNA:RNA hybrid accumulation and hybrid-dependent fork slowing in human cells—that is, estrogen-dependent transcriptional burst[42], TOP1 depletion[33], and TOP1 inhibition[47]—we observed accumulation of nascent strand discontinuities, persisting well after replication. Combined, this evidence strongly suggests that post-replicative DNA:RNA hybrids are frequent and pathological intermediates at TRCs and may contribute to the observed impairment of fork progression. We envision three potential mechanisms through which post-replicative DNA:RNA hybrids can be generated (Extended Data Fig. 8a). First, transcription of a hybrid-prone sequence could be initiated behind the replication fork, and the elongating RNA may anneal to the unwound lagging strand template, as has recently been suggested[27]. Alternatively, the nascent RNA could invade the duplex daughter DNA strand, forming a post-replicative R-loop that is nucleolytically processed into a hybrid (Extended Data Fig. 8a, 1). Second, a hybrid ahead of the fork could remain bound and subsequently bypassed by the replisome (Extended Data Fig. 8a, 2). Finally, the annealed mRNA ahead of the fork may be displaced by replicative and/or accessory helicases and reanneal behind the replication fork (Extended Data Fig. 8a, 3). Because nascent chromatin is transcriptionally silenced[52], we consider it unlikely—at least in the eukaryotic system—that the RNA is newly synthesized behind the fork, and currently favor the hypothesis that RNA might be inherited from preexisting hybrids ahead of the fork. The replicative helicase activity—which differs profoundly between prokaryotes and eukaryotes—may determine to what extent the hybrid remains intact; although it has been suggested that the RNA remains bound during prokaryotic TRCs[25], it appears to be liberated during eukaryotic TRCs[26]. Interestingly, we observed comparable phenotypes (fork slowing and post-replicative hybrid formation) in both systems, suggesting that the impact of the bound RNA is independent of its origin. Whether post-replicative DNA:RNA hybrid formation requires specific RNA helicases and/or active transcription during replication will be fascinating questions for future studies. DNA:RNA hybrid resolving enzymes, such as SETX and RNase H2 (refs. [53,54]), have been shown to travel with the replisome, suggesting that—in physiological conditions—DNA:RNA hybrids forming on the lagging strand are efficiently removed to allow unperturbed replication progression (Fig. 6). However, in case of excessive hybrid formation—for example, multiple or particularly long hybrids, as we observed upon transcriptional bursts or RNase H inactivation (Fig. 6)—these processing mechanisms might be insufficient or saturated. In that case, 'postponing' hybrid processing may be favorable, to enable restoration of fork progression and to limit excessive ssDNA accumulation, leading to post-replicative hybrids that remain embedded within the daughter duplexes long after fork passage. Intriguingly, several factors involved in fork restart and DNA-damage tolerance behind replication forks have recently been shown to contribute to hybrid resolution, through yet-elusive mechanisms[27,28]. Further investigations will be needed to address whether and how specific enzymes involved in DNA:RNA hybrid metabolism—some of which were implicated in human disease[17,55–60]—are specifically involved in resolving or preventing post-replicative DNA:RNA hybrids. Moreover, analogous analyses on co-directional conflicts may help clarify why—across species—head-on conflicts have more deleterious consequences than do co-directional ones[9,10].

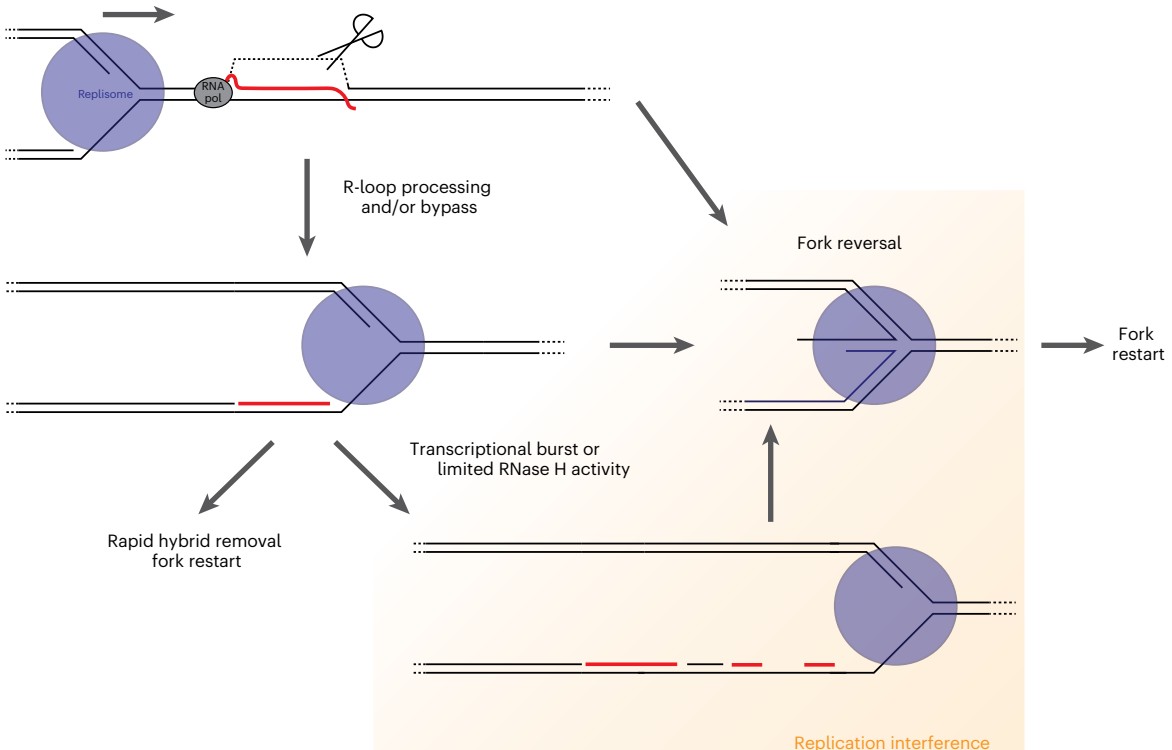

**Fig. 6 | Working model for the accumulation of post-replicative DNA:RNA hybrids at TRCs and their impact on fork progression.** During head-on transcription replication conflicts, R-loops can form behind the RNAP and are either processed into DNA:RNA hybrids and/or are efficiently bypassed by the replisome. The DNA:RNA hybrid is thereby transferred to the lagging strand behind the replisome (see Extended Data Figure 8a for potential mechanisms). Under physiological conditions, this hybrid is rapidly resolved, allowing unperturbed fork progression. In case of excessive accumulation (that is transcriptional burst) or impaired removal (that is limited RNase H activity), DNA:RNA hybrids may be still efficiently bypassed, but their accumulation behind the fork would result in delayed fork progression and frequent fork reversal (see Extended Data Fig. 8b for potential mechanisms). Upon stress resolution, the reversed fork can be restarted to complete DNA replication.

## DNA:RNA hybrids, topological stress, and fork remodeling

We observed that excessive DNA:RNA hybrid accumulation at replication forks is associated with signs of replication stress, such as fork slowing and replication fork reversal. DNA:RNA hybrid accumulation behind the fork may impair replication fork progression through several different mechanisms (Extended Data Fig. 8b). First, unresolved DNA:RNA hybrids might remain tethered to the replisome to await processing by replisome-associated RNA nucleases or helicases (Extended Data Fig. 8b, 1). This could impose excessive DNA looping at ongoing forks, increase torsional constraints, and finally impair replisome function. Topological constraints have previously been linked to replication interference and genomic instability at TRCs, although they are mainly attributed to the build-up of positive supercoiling in between replication and transcription machinery on parental DNA[3,32]. Sterical exclusion of topoisomerases and/or excessive supercoiling ahead of the fork may impede resolution of torsional constraints, which may, however, be transferred behind the fork through fork rotation, leading to pre-catenane formation[31]. Importantly, excessive fork rotation per se may interfere with fork progression and prevent efficient post-replicative processing of DNA:RNA hybrids[61], further exacerbating DNA:RNA hybrid accumulation. The topological constraints behind the replication fork might therefore be a combination of both the topological stress directly arising behind the fork and topological stress being transferred from ahead of the fork. Accordingly, bacterial topoisomerases have recently been shown to work both ahead and behind replication forks at TRCs[34]. Second, hybrid processing can expose stretches of excessive ssDNA, leading to RAD51 loading and fork reversal and thereby transiently preventing replication fork progression (Extended Data Fig. 8b, 2)[62]. Indeed, in line with recent evidence on drug-induced accumulation of DNA:RNA hybrids[47], fork reversal emerged in our study as a critical response to endogenous transcription-replication conflicts. Besides ssDNA accumulation, increased topological stress[63] was shown to promote fork remodeling (Fig. 6 and Extended Data Fig. 8b). The direct causative link between DNA:RNA hybrid formation and fork reversal, as well as the interplay with other processes, such as repriming, remain to be explored. Moreover, besides affecting fork remodeling and topology, accumulation of DNA:RNA hybrids may interfere with efficient chromatinization of the replicated duplexes (Extended Data Fig. 8b, 3), which has previously been shown to alter the rate of replication fork progression[64]. Finally, fork reversal has previously been shown to extend as a global nuclear response, even though lesions may locally affect only a small number of replication forks[65]. It is thus conceivable that—despite the relatively low number of DNA:RNA hybrids directly detected by our EM methods—fork remodeling may extend to a much higher number of replication forks than those directly challenged by hybrids and hybrid-associated topological constraints or chromatin-maturation issues.

To conclude, our study highlights post-replicative DNA:RNA hybrids as the most abundant intermediates accumulating at transcriptionally challenged forks, raising a plethora of new and intriguing questions to be explored. This may motivate researchers to revisit previous observations and orient future work, to expand our understanding of how formation of DNA:RNA hybrids interferes with ongoing replication and ultimately drives genomic instability.

## Online content

Any methods, additional references, Nature Portfolio reporting summaries, source data, extended data, supplementary information,

acknowledgements, peer review information; details of author contributions and competing interests; and statements of data and code availability are available at https://doi.org/10.1038/s41594-023-00928-6.

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

## Methods

### Cell lines

Human MCF7 cells (gift from K. Cimprich), HeLa cells (gift from P. Pasero), and U2OS (ATCC, HTB-96) cells were cultured in Dulbecco's modified Eagle's medium (DMEM) supplemented with 10% fetal bovine serum (FBS), 100 U/ml penicillin and 100 mg/ml streptomycin in an atmosphere containing 6% $CO_2$ at 37 °C. For estrogen stimulation, MCF7 cells were allowed to settle at low confluency in normal growth medium for >16 hours, were washed 3× with warm PBS, kept in arrest medium (phenol-red free DMEM, 10% charcoal-stripped FBS) for 48 hours and subsequently released in fresh arrest medium with 100 nM estradiol for 2 hours, unless otherwise indicated. For TOP1 downregulation, HeLa control and shTOP1 cells were cultured with 2 µg/ml doxycycline for 72 hours.

### Bacterial strains

Both *B. subtilis* strains (HM1300 and HM2043) used in this study were previously constructed[9]. Strains were streaked on LB agar plates and supplemented with spectinomycin (100 µg/mL). Precultures were inoculated from single colonies into 5 mL of LB broth and incubated at 37 °C with shaking. Precultures were diluted to an optical density at 600 nm ($OD_{600}$) of 0.05 and used to inoculate 10 mL experimental cultures, which were grown at 30 °C with shaking. After 90 minutes, IPTG was added to a final concentration of 1 mM (for induced cultures). All cultures were grown until the $OD_{600}$ was ~0.3, and sodium azide was added to a final concentration of 0.02% to fix the cultures.

### In vitro transcription for R-loop formation

pFC53 plasmid was transcribed with 20 U/µg of T3 RNA polymerase in 1× transcription buffer, supplemented with 20 mM DTT, 0.05% Tween-20 and 50 µM rNTP, at 37 °C for 45 minutes and subsequently inactivated at 65 °C for 10 minutes. For the non-transcribed control, the T3 RNA polymerase was omitted. All samples were digested with 15 µg/mL RNase A and 1.25 mg/mL proteinase K at 37 °C for 30 minutes each. DNA was cross-linked with 10 µg/mL 4,5',8-trimethylpsoralen, followed by a 3-minute irradiation pulse with ultraviolet (UV) 365 nm monochromatic light (UV Stratalinker 1800; Agilent Technologies). DNA was purified using chloroform/isoamylalcohol (24:1) and precipitated with 2 volumes of 100% ethanol and 0.1 M sodium acetate at −20 °C overnight. Finally, the DNA was washed with 70% ethanol, briefly dried at 37 °C and resuspended in $dH_2O$. For linearization, transcribed plasmid was digested with 40 U/µg XmnI for 3 hours at 37 °C. RNase H controls were additionally digested with 10 U/µg RNase H at 37 °C overnight. DNA was purified using the Silica Bead DNA Gel Extraction Kit, according to manufacturer's instructions.

For gel shift mobility assay, 150 ng of plasmid was run on a 0.9% agarose gel in 1× TBE without intercalating agent.

### Dot blot

Digested DNA (either pFC53 or MCF7 genomic DNA) was diluted in 2× SSC (saline sodium citrate buffer), split in two, and spotted onto a positively charged Zeta probe membrane. Once dried, the membrane was cross-linked in a hybridization oven at 80 °C for 1 hour and blocked in 5% milk/PBS/0.1% Tween-20 at room temperature (RT) for 1 hour. Membranes were incubated with primary antibodies (Kerafast S9.6, 0.5 µg/ml or anti-dsDNA 1:1000 in 3% BSA/PBS/0.1% Tween-20) at 4 °C overnight. Membranes were washed three times with PBS/0.1% Tween-20 and incubated with secondary antibody (anti-mouse-HRP 1:5,000 in 1.5% BSA/PBS/0.1% Tween-20) at RT for 1 hour. Membranes were developed with ECL and imaged on Fusion-Capt Advance Solo7 imaging system (Vilber Lourmat).

### Transfections

For RNase H1-GFP overexpression, MCF7 cells were transfected with pAIO RNH1-GFP using Lipofectamine 3000, according to

manufacturer's instructions, 48 hours prior to estrogen stimulation. For ZRANB3 downregulation, MCF7 cells were transfected with siRNA targeted against ZRANB3 (40 nM) using RNAiMAX, according to the manufacturer's instructions, 96 hours prior to estrogen stimulation.

### Immunoblotting

Whole-cell extracts of MCF7 cells were prepared in Laemmli sample buffer (4% SDS, 20% glycerol, and 120 mM Tris- HCl, pH 6.8), loaded onto 4%–20% Mini-PROTEAN TGX Precast Protein Gels and separated by electrophoresis at 160 V at RT. Proteins were transferred to Immobilon-P membranes for 70 minutes at 100 V in ice-cold transfer buffer (25 mM Tris, 192 mM glycine, 20% methanol). Membranes were blocked in either 5% milk/TBS/0.1% Tween-20 (for actin detection) or 2% ECL blocking solution (for ZRANB3 detection) at RT for 1 hour and incubated with primary antibodies (anti-ZRANB3 1:1,000 in 2% ECL blocking solution, anti-actin 1:2,000 in 5% milk/TBS/0.1% Tween-20) at 4 °C overnight. Membranes were washed three times with TBS/0.1% Tween-20 and incubated with secondary antibody (anti-mouse-HRP or anti-rabbit-HRP 1:5,000 in 1.5% BSA/PBS/0.1% Tween-20) at RT for 1 hour. Membranes were developed with ECL and imaged on Fusion-Capt Advance Solo7 imaging system (Vilber Lourmat).

### DNA fiber spreading analysis

Following estrogen depletion and stimulation, MCF7 cells were sequentially pulse-labeled with 30 µM CldU and 250 µM IdU for 30 minutes. The cells were collected by scraping in ice-cold PBS, washed, and resuspended in PBS at $3 \times 10^5$ cells/mL. Then, 3 µL of cells was mixed with 7 µlL of lysis buffer (200 mM Tris-HCl pH 7.5, 50 mM EDTA, 0.5% SDS) on a glass slide. After 5 minutes, the slides were tilted at 15–45°, and the resulting DNA spreads were air dried, fixed in 3:1 methanol/acetic acid overnight at 4 °C. The fibers were denatured with 2.5 M HCl for 1 hour, washed with PBS and blocked in 2% BSA/PBS/0.1% Tween for 40 minutes. The newly replicated CldU and IdU tracks were labeled with anti-BrdU antibodies recognizing CldU (1:500) and IdU (1:100) for 2.5 hours at RT, followed by 1 hour of incubation with secondary antibodies (anti-mouse-AlexaFluor 488, 1:300, and anti-rat-Cy3, 1:150) at RT in the dark. Fibers were mounted using ProLong Gold AntiFade, visualized (Leica DMI 6000; objective lenses: HC PL APO ×63, 1.40 numerical aperture (NA) oil; Leica Application Suite X 3.6.0.20104) and analyzed using ImageJ software (version 2.0.0-rc-43/1.51h). Of note, we noticed that MCF7 cells experience issues incorporating IdU and therefore considered the CldU tract length as the more reliable readout for these fiber experiments.

### Flow cytometry

For flow cytometry analysis, MCF7 cells were labeled with 10 µM EdU for 30 minutes, collected by scraping into ice-cold PBS, and fixed with 4% PFA/PBS for 15 minutes at RT. In between all the following steps, cells were washed twice with 1% BSA/PBS at 500$g$ for 5 minutes. For γH2AX detection, cells were incubated with primary antibody (ms anti-γH2AX, 1:1,000) in 1× saponin buffer for 2 hours at RT with occasional inversions and subsequently with secondary antibody (anti-mouse-Alexa Fluor 647, 1:125) for 30 minutes at RT. For all experiments, the EdU Click reaction was performed for 30 minutes at RT, according to the manufacturer's instructions. Finally, cells were resuspended in 1% BSA/PBS, containing 0.1 mg/ml RNase A and 1 µg/ml DAPI. Samples were measured on an Attune NXT flow cytometer (Beckman Coulter; Attune NTX Software Version 4.2.0.) and analyzed by the FlowJo software (version 10.4).

### EdU alkaline comet assay

MCF7, U2OS or HeLa cells were treated as indicated and labeled with 10 µM EdU for the final 30 minutes of the treatment. Cells were washed with PBS and incubated in fresh medium without EdU for variable time periods (chase time of 0 hours, 0.5 hours, 1 hour, or 1.5 hours); the respective treatments were maintained during the entire chase time.

Cells were then collected by trypsinization, embedded in 0.8% Sea-Plaque low-melting point agarose on two-well comet slides, and lysed in lysis buffer (10 mM Tris pH 10, 2.5 M NaCl, 100 mM EDTA, 10% DMSO, 1% Triton X-100). Slides were then washed in PBS and incubated in denaturation buffer (300 mM NaOH, 1 mM EDTA) for 40 minutes, followed by electrophoresis for 20 minutes at 18 V and 300 mA. Afterwards, slides were washed in PBS, fixed in ice-cold ethanol for 10 minutes, and dried at 37 °C. Slides were then subjected to EdU Click IT reaction for 30 minutes at RT, washed with PBS, stained with SYBR for 15 minutes at RT, and washed again. The dried slides were imaged (Leica DMI 6000; objective lenses: HC PL FLUOTAR ×40/0.80) and analyzed for their olive tail moment using the Open Comet plugin (version 1.3.1) for ImageJ (version 2.0.0-rc-43/1.51h). To assess nascent strand maturation over time, median olive tail moments were normalized to the 0-hour chase time point. Half-lives of nascent strand discontinuities were calculated by fitting an exponential decay function to the normalized olive tail moments.

## Conflict enrichment by gel electrophoresis

Fixed *B. subtilis* cells were collected by centrifugation at 5,500g for 10 minutes and washed with ice-cold PBS three times. Cells were then resuspended in PBS and cross-linked with 200 µg/mL 4,5′,8-trimethylpsoralen, followed by a 2-minute irradiation pulse with UV 365 nm monochromatic light. Cross-linked cells were washed with ice-cold PBS three times and subsequently resuspended in 0.5% low melt NuSieve GTG agarose, containing 125 µg/mL lysozyme (dissolved in water). The mixture was placed in an agarose plug mold until solidified. Agarose plugs were then incubated in 1 mL of 10 mM Tris (pH 7.5) 500 mM EDTA buffer at 37 °C overnight. After incubation, 400 µL of 5% sarkosyl 500 mM EDTA buffer and 100 µL of 20 mg/mL proteinase K was added. The plugs were then incubated at 50 °C for 5 hours and subsequently washed eight times (for 4 hours each wash step) with 2 mM Tris (pH 8) 1 mM EDTA buffer at 4 °C. For digestion, agarose plugs were equilibrated in 1× cutsmart buffer for at least 4 hours at RT and 1 hour at 37 °C and then digested with 180U BstBI, 180U NruI-HF, 180U PstPI and 90U BfaI in a wet chamber at 37 °C overnight. The next day the digestion mixture was renewed for an additional 3hrs at 37 °C. Plugs were poured into a 0.5% low-melting NuSieve GTG agarose gel in 1× TBE and run at 50 V for 21 hours at 4 °C without intercalating agents. The agarose gel was stained with ethidium bromide, and areas of interest excised with minimal UV exposure for gel extraction using the Silica Bead DNA Gel Extraction Kit according to manufacturer's instructions with minor modifications: Gel pieces were weighed and dissolved in 0.5 volumes/weight of TBE conversion buffer and 2 volumes/weight of binding buffer for a maximum of 5 minutes at 50 °C. Silica beads were added according to DNA content predictions, incubated at 50 °C for 5 minutes to allow DNA binding, and washed three times briefly with ice-cold washing buffer. During washing steps, the pellet was not resuspended to prevent DNA shearing. The beads were air dried for 10 minutes, and the DNA was eluted 2–4 times in TE pH 7.4 or dH$_2$O; the volume for each elution was chosen to be equal to the volume of beads used in the reaction. If gel pieces exceeded the recommended volumes, the reaction was split into separate tubes after the gel was dissolved. In that case, eluates were pooled and subsequently concentrated by speed vac. A detailed protocol is available in ref. [49].

## MCF7 genomic DNA extraction and digestion

Following estrogen depletion and stimulation, MCF7 cells were collected by scraping in ice-cold PBS, washed, and lysed in TE buffer pH 7.4, containing 0.5% SDS and 0.6 mg/mL proteinase K at 37 °C overnight. The next day, 1.8 volumes of G2 buffer (800 mM guanidine-HCl, 30 mM Tris-HCl (pH 8.0), 30 mM EDTA (pH 8.0), 5% Tween-20, and 0.5% Triton X-100), containing 0.3 mg/ml proteinase K were carefully added to the samples and incubated at 37 °C for 1.5 hours. The DNA was subsequently purified using chloroform/isoamylalcohol (24:1)

and precipitated in two volumes of 100% ethanol and 0.1 M sodium acetate. Finally, the DNA was washed with 70% ethanol, briefly dried at 37 °C, and resuspended in TE buffer pH 7.4 at 37 °C with slight agitation.

For dot blot and EM analysis, 10 µg of this genomic DNA was digested with the indicated restriction enzymes at 37 °C overnight. Samples were then digested with 0.25 mg/mL RNase A, 25 U/mL RNase T1, 25 U/mL RNase III ± 500 U/mL RNase H in NEB RNase H buffer, supplemented with 0.5 mM NaCl for 2.5 hours at 37 °C. The digested DNA was purified using the Silica Bead DNA Gel Extraction Kit, according to the manufacturer's instructions.

## DNA:RNA immunoprecipitation

Extracted and purified genomic DNA from MCF7 cells was digested with HindIII-HF, EcoRI, BsrGI, XbaI, and SspI at 37 °C overnight, purified with phenol/chloroform/isoamylalcohol (25:24:1) and precipitated with 2 volumes 100% ethanol, 0.1 mM sodium acetate, and 75 µg/ml glycogen at −20 °C overnight. Finally, the DNA was washed with 70% ethanol, briefly dried at 37 °C and resuspended in TE buffer pH 7.4 at 37 °C with slight agitation. RNase H controls were digested with 10U/µg RNase H at 37 °C overnight. For the DNA:RNA immunoprecipitation (DRIP), 4.4 µg of digested and purified DNA was incubated with 10 µg of S9.6 antibody (Kerafast) in a total of 500 µL 1× binding buffer (100 mM NaPO$_4$ pH 7.0, 1.4 M NaCl, 0.5% Triton X-100) at 4 °C overnight with steady rotation. Ten percent of the sample was kept aside as input control prior to antibody addition. To immobilize antibody-bound material, samples were incubated with 50 µL of magnetic protein A/G beads at 4 °C for 2 hours with steady rotation. Beads were washed three times for 10 minutes with 1× binding buffer and incubated in 250 µL 1× elution buffer (50 mM Tris-HCL pH 8.0, 10 mM EDTA pH 8.0, 0.5% SDS) with 0.6 mM proteinase K at 55 °C for 45 minutes with occasional inversion. Eluted DNA was purified by phenol/chloroform/isoamylalcohol (25:24:1) extraction and ethanol/sodium acetate/glycogen precipitation. qPCR was performed on a Roche LightCycler 480 Instrument II using SYBR-Green master mix (Bio-Rad Laboratories). Primers used for qPCR are listed in the Supplementary Table.

For *B. subtilis*, digested genomic DNA was extracted from whole plugs using the Silica Bead DNA Gel Extraction Kit, as described above. RNase H controls were digested with 2.5 U/µg RNase H at 37 °C overnight. DRIP was performed as described above, with minor changes (500 ng of material, 5 µg of S9.6 antibody, 25 µg of beads).

## Native electron microscopy

For native DNA spreading for EM analysis, 50–150 ng of DNA was mixed with benzyldimethylalkylammonium chloride (BAC) and formamide, spread on a water surface and loaded on carbon-coated 400-mesh copper grids, as previously described[39]. Subsequently, DNA was coated with platinum using the high vacuum evaporator MED 020 (BalTec). In vitro transcription samples and *B. subtilis* samples were imaged manually using a Tecnai G2 Spirit transmission electron microscope (FEI; LaB6 filament; high tension ≤120 kV; acquisition software: DigitalMicrograph Version 1.83.842 (Gatan)) with a side-mounted digital camera Gatan Orius 1000 (2,600 × 4,000 pixels). For MCF7 samples, imaging was automated using a Talos 120 transmission electron microscope (FEI; LaB6 filament, high tension ≤120 kV) with a bottom mounted CMOS camera BM-Ceta (4,000 × 4,000 pixels) and the MAPS software package (version 3.16, Thermo Fisher Scientific). Samples were annotated for molecules of interest using the MAPS Viewer software (version 3.16) and overlapping images for annotated regions were stitched together using the automated pipeline ForkStitcher (https://github.com/jluethi/ForkStitcher, Version 0.1.1), developed by J. Lüthi. The stitching was based on the ImageJ Stitching plugin (ref. [66], version 3.1.6), used through the PyImageJ library. Total DNA content was determined using a DNA-content quantification algorithm (https://github.com/roessler-f/DNAQuantification), developed by F. Roessler. A detailed protocol for both pipelines is available in ref. [43].

## S9.6-gold electron microscopy

For in vitro-transcribed material, 80–100 ng of transcribed and purified pFC53 was labeled with 10 µg/mL S9.6-gold NPS conjugate for 1 hour at 37 °C, cross-linked with 0.2% glutaraldehyde for exactly 15 minutes at 37 °C. For *B. subtilis*, gel-extracted material was mixed with 80–100 ng of transcribed and purified pFC53 prior to S9.6 labeling. When indicated, these samples were digested with 10 U RNase H overnight at 37 °C and purified using the Silica Bead DNA Gel Extraction Kit, according to the manufacturer's instructions. *B. subtilis*/pFC53 samples were then incubated with 10 µg/mL S9.6-gold conjugate for 1 hour at 37 °C, cross-linked with 0.2% glutaraldehyde for exactly 15 minutes at 37 °C.

In either case, the labeled DNA was immediately mixed with benzyldimethylalkylammonium chloride (BAC) and formamide, spread on a water surface and loaded on carbon-coated 400-mesh copper grids. Subsequently, DNA was coated with platinum using a high vacuum evaporator MED 020 (BalTec). Microscopy was performed manually with using a Tecnai G2 Spirit transmission electron microscope (FEI; LaB6 filament; high tension ≤120 kV; acquisition software: Digital-Micrograph Version 1.83.842 (Gatan)), with a side-mounted digital camera Gatan Orius 1000 (2,600 × 4,000 pixels). For each experimental condition, at least 70 molecules were analyzed using ImageJ (National Institutes of Health, version 2.0.0-rc-43/1.51h). A detailed protocol is available in ref. [49].

## Quantification and statistical analyses

Statistical analysis was performed using GraphPad Prism 8 (version 8.4.2). Data normality was tested prior to statistical analysis, and the statistical test used for each experiment is stated in the corresponding figure legend. Every experiment was repeated at least twice. Every EM analysis was based on at least 70 individual molecules per condition, every DNA fiber experiment on 100 individual fiber tracts per condition, and every comet assay at least on 30 individual cells.

## Reporting summary

Further information on research design is available in the Nature Portfolio Reporting Summary linked to this article.

## Data availability

Source data are provided with this paper. The remaining original microscopy images exceed several terabytes and will be made available upon reasonable request.

## Code availability

The code for fork stitching is available at: https://github.com/jluethi/ForkStitcher, version 0.1.1.
The code for the DNA content quantification algorithm can be found at: https://github.com/roessler-f/DNAQuantification.

## References

66. Preibisch, S., Saalfeld, S. & Tomancak, P. Globally optimal stitching of tiled 3D microscopic image acquisitions. *Bioinformatics* **25**, 1463–1465 (2009).

## Acknowledgements

We are grateful to J. Lüthi, F. Roessler, and the Center for Microscopy and Image Analysis of the University of Zurich for assistance with microscopy and imaging analysis. We thank S. Hamperl for technical assistance and critical reading of the manuscript, and members of the Lopes lab for useful discussions and suggestions on the manuscript. Work in the Lopes lab was supported by the SNF grants 31003A_169959 and 310030_189206, and by support of the Forschungskredit UZH to H. S.. K. A. C. is an ACS Research Professor and is supported by the NIH grant R01GM119334, while work in the Merrikh lab was supported by the NIH grant R01GM128191.

## Author contributions

H. S., K. Z., K. A. C., H. M. and M. L. conceived the project and designed experiments. K. Z. performed key set-up experiments in the initial phase of this project. H. S. prepared all samples, with the help of K. S. L. for *B. subtilis*. H. S. performed all experiments. H. S. analyzed all EM data and was assisted by J. K. and J. A. S. H. S. and D. K. performed DNA fiber assays and comet assays with the help of J. K. M. P. C. assisted with qPCR–DRIP and comet analysis. M. L. supervised the project. H. S. and M. S. prepared and revised the manuscript with input from all co-authors. Any correspondence should be addressed to M. L.

## Funding

## Competing interests

The authors declare no competing interests.

## Additional information

**Extended data** is available for this paper at https://doi.org/10.1038/s41594-023-00928-6.

**Correspondence and requests for materials** should be addressed to Massimo Lopes.

**Peer review information** *Nature Structural & Molecular Biology* thanks Andrés Aguilera and the other, anonymous, reviewer(s) for their contribution to the peer review of this work. Carolina Perdigoto, Beth Moorefield, Tiago Faial, and Dimitris Typas were the primary editors on this article and managed its editorial process and peer review in collaboration with the rest of the editorial team. Peer reviewer reports are available.

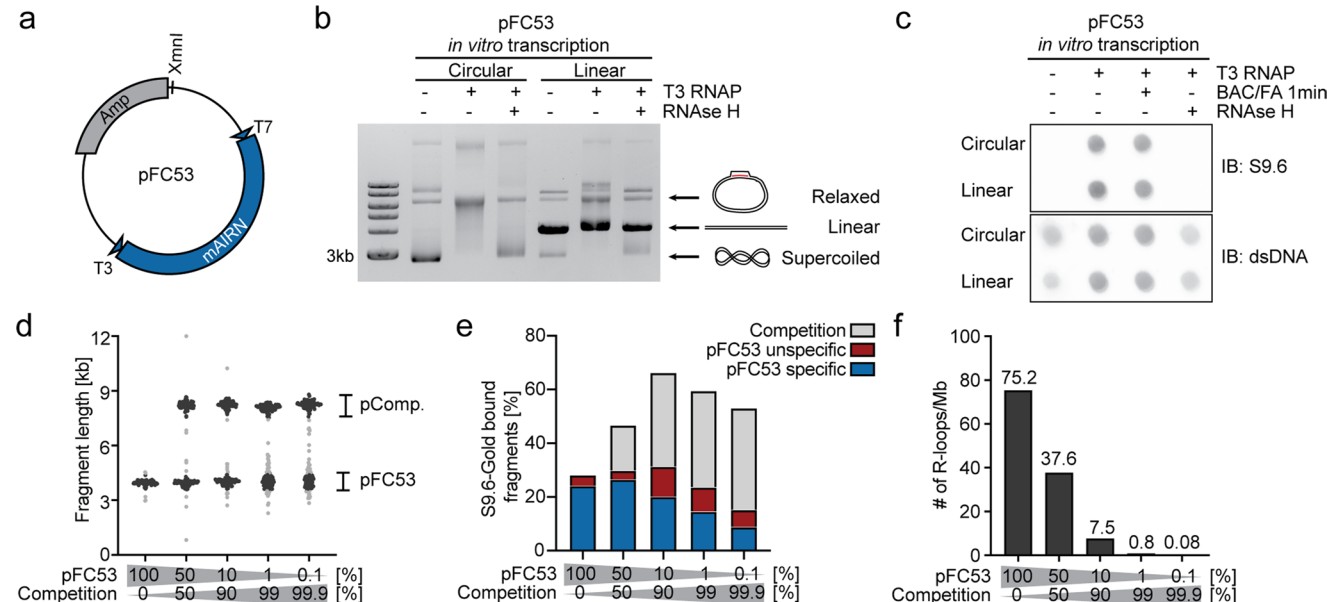

**Extended Data Fig. 1 | Direct visualization of in vitro-generated R-loops using Electron Microscopy (EM). a)** pFC53 map including the R-loop prone *mAIRN* gene, the T3 promotor used in this study and the XmnI linearization site. **b)** Gel shift assay of circular and linear pFC53 +/− RNase H treatment. This result has been reproduced 3 independent times. **c)** Dot blot of circular and linear pFC53 +/− RNase H, immunoblotted for S9.6 and dsDNA (as loading control). When indicated, pFC53 was incubated with 20% formamide and 0.02% BAC for 1 min prior to dot blot loading. This result has been reproduced 3 independent times. **d-f)** Competition experiment to assess the specificity and selectivity of the S9.6-gold conjugated antibody. The linearized competition plasmid and

R-loop carrying pFC53 were mixed in the indicated ratios, labeled with S9.6-gold and spread for EM analysis. For each plasmid at least 100 molecules were analyzed. The experiment was reproduced once. **d)** Length distribution of the imaged fragments. pFC53 and the competition plasmid were identified by their respective sizes. The relative frequencies of pFC53 and competition plasmid displayed do not represent the frequencies observed in the sample. **e)** Quantification of S9.6-gold binding to the pFC53 and the competition plasmid. Binding to pFC53 was further differentiated into specific (within the *mAIRN* gene) or unspecific (outside of the *mAIRN* gene). **f)** Calculated R-loop density (number of R-loops/Mb) for the ratios indicated below.

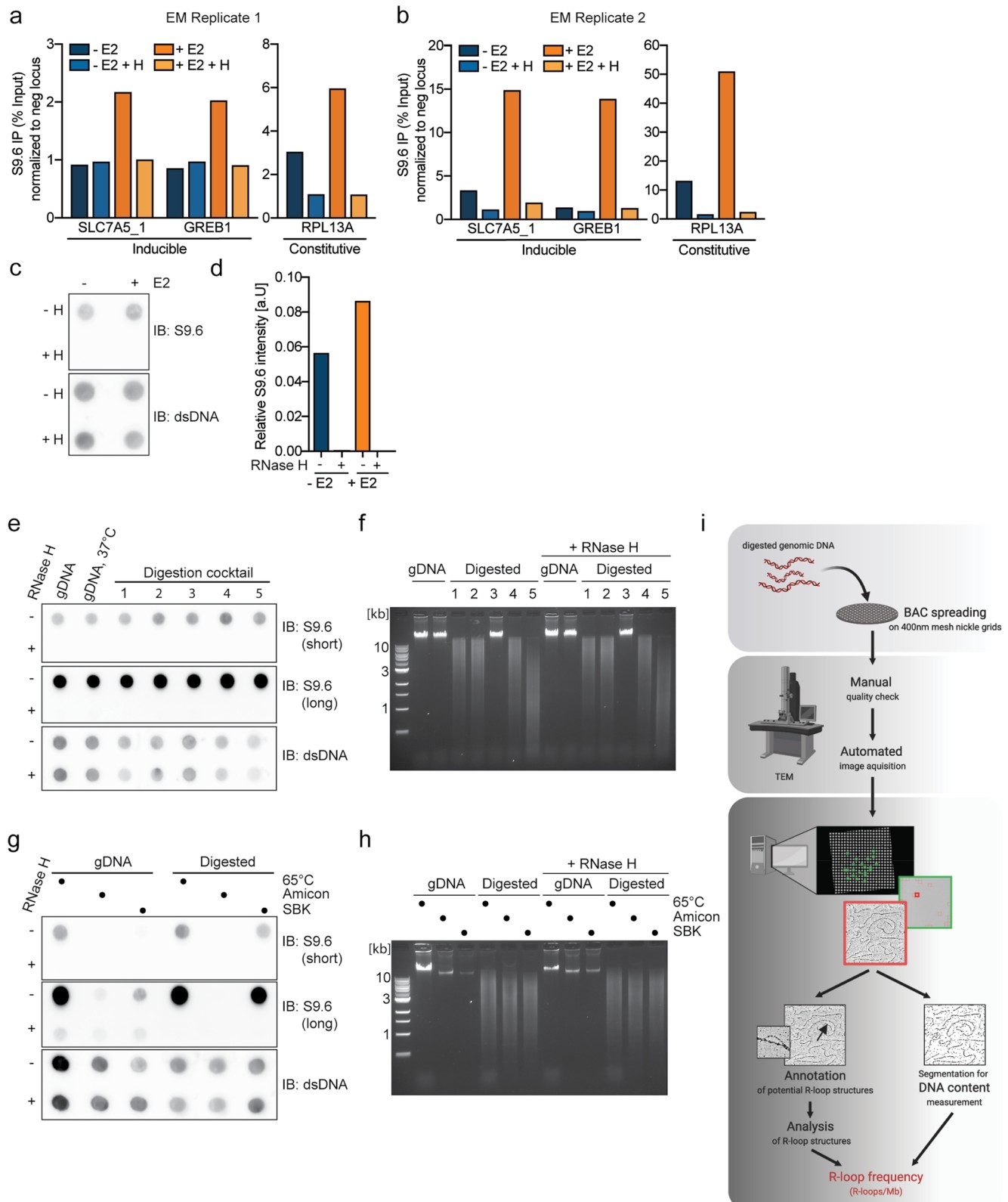

**Extended Data Fig. 2 | See next page for caption.**

**Extended Data Fig. 2 | Optimization of EM sample preparation to detect R-loop structures on genomic DNA. a–b)** qPCR-DRIP analysis of MCF7 cells +/− 2 h E2 +/− in vitro RNase H digestion for representative E2-inducible and constitutive genomic loci on the extracted genomic DNA used for the two biological replicates of the EM analysis. Data was normalized to a negative locus. a) corresponds to data shown in Fig. 2b-g, b) corresponds to data shown in Fig. 2e–g, Extended Fig. 2c, d and Extended Data Fig. 3a, b. **c)** Dot blot analysis of genomic DNA extracted from MCF7 +/− 2 h E2 +/− in vitro RNase H digestion. Genome wide hybrid accumulation was detected by S9.6 immunostaining (loading control: dsDNA). **d)** Quantification of integrated intensities in b); S9.6 signal was normalized to the respective dsDNA loading control. **e)** Genomic DNA, extracted from E2 stimulated MCF7, was digested with different cocktails of restriction enzymes: 1. PvuII, 2. PvuII + EcoRI, 3. NotI, 4. BbvCI, 5. HindIII + EcoRI + BsrGI + XbaI + SspI. Hybrid levels were detected by S9.6 dot blot (S9.6 blot shown as short and long exposure; loading control: dsDNA). Digestion cocktail 2 was used for all follow-up experiments. This result was reproduced once. **f)** Agarose gel electrophoresis of samples in e) showing the different degrees of genomic DNA fragmentation. **g)** Dot blot of extracted genomic DNA, extracted from E2 stimulated MCF7 and purified either directly or after restriction enzyme digest using either size exclusion columns (Amicon) or a silica bead extraction kit (SBK). Note that, for unknown reasons, hybrids are not recovered from Amicon columns, which is why SBK was selected for all follow-up experiments. Heat inactivation was used as positive control for hybrid stability. Hybrid stability was assessed by S9.6 dot blot (S9.6 blot shown as short and long exposure; loading control: dsDNA). This result was reproduced once. **h)** Agarose gel electrophoresis of samples in g) showing the degree of genomic DNA fragmentation. **i)** Automated high-throughput EM workflow used to image and quantify R-loops on human genomic DNA[43].

a

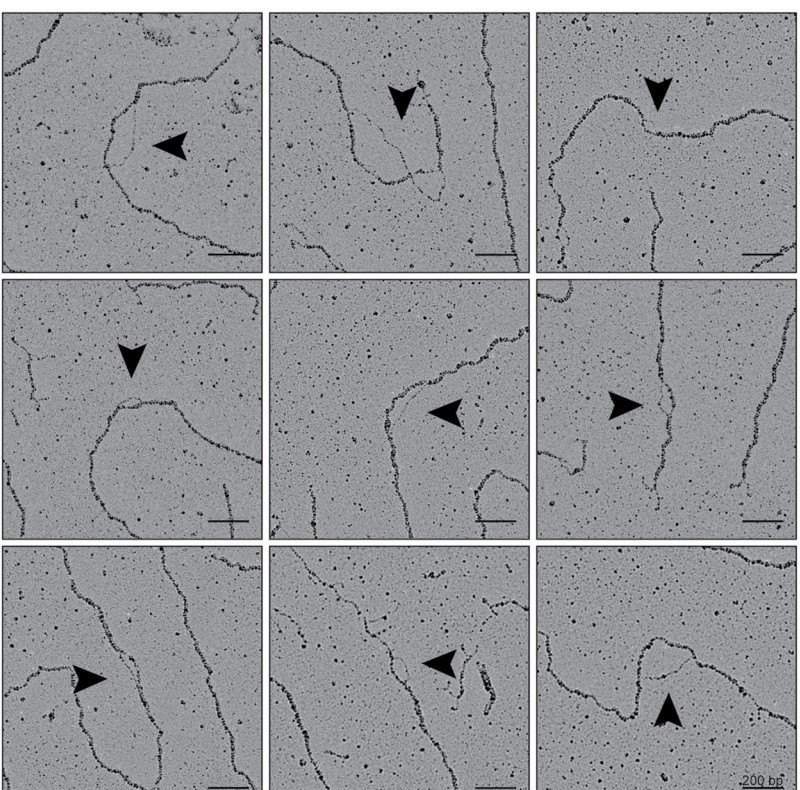

b

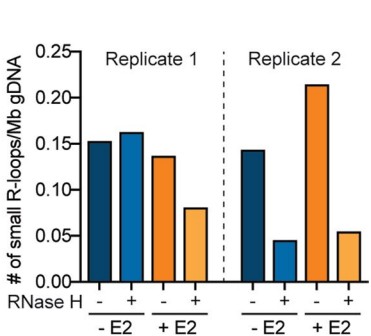

**Extended Data Fig. 3 | EM-based visualization and quantification of R-loops on human genomic DNA upon estrogen-dependent transcriptional burst.** **a)** Representative electron micrographs of R-loops found on genomic DNA from E2 stimulated cells. Scale bar: 200 bp/72 nm. **b)** Frequency of R-loops that are <300 bp in size in two independent biological replicates. Absolute numbers of R-loops <300 bp were normalized to the total DNA content within the analyzed area.

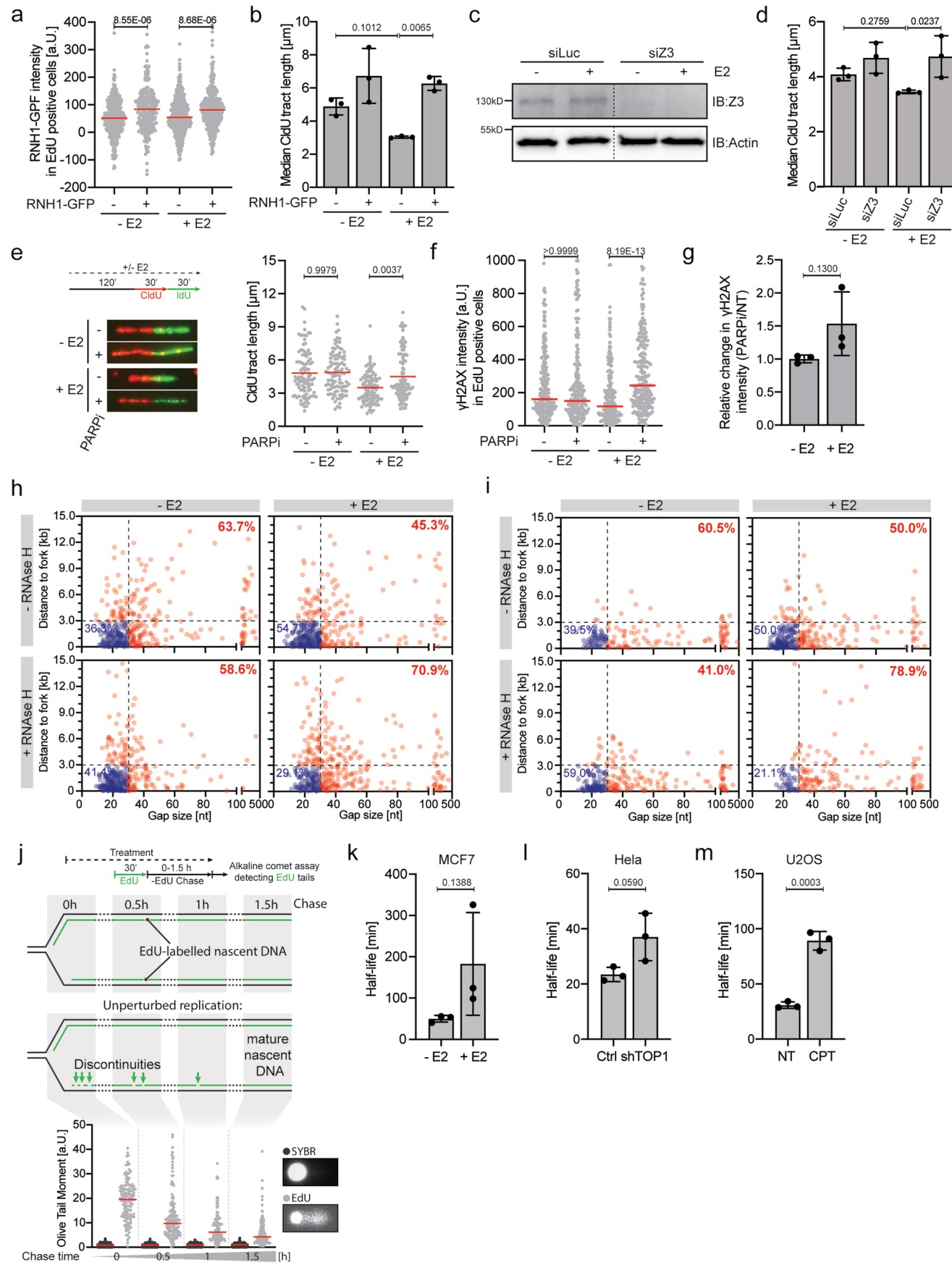

**Extended Data Fig. 4 | See next page for caption.**

**Extended Data Fig. 4 | Estrogen-dependent transcriptional burst results in replication stress and is associated with hybrid accumulation behind the replication fork. a)** Single cell GFP values as measured by FACS of MCF7 +/− E2 +/− 48 h RNH1-GFP overexpression. Red line: median. Statistical significance determined by Kruskal-Wallis test, followed by Dunn test. **b)** Median CldU tract length from three independent biological replicates of Fig. 3a), shown as mean +/− s.d..Statistical significance determined by ordinary one-way ANOVA, followed by Sidak test. **c)** Immunoblot detection of indicated proteins in whole cell extracts from MCF7 cells +/− E2 +/− 96 h siRNA transfection. Dotted line indicates where samples unrelated to the experiment have been omitted. **d)** Median CldU tract length from three independent biological replicates of Fig. 3b), shown as mean +/−s.d. Statistical significance determined by ordinary one-way ANOVA, followed by Sidak test. **e)** DNA fiber assay of MCF7 +/− E2. Left: assay setup with representative DNA fiber images. Right: quantification of CldU tract lengths [μm]; at least 100 individual molecules quantified per conditions. Red line: mean. Statistical significance determined by Kruskal-Wallis

test, followed by Dunn test. **f)** Single cell gH2AX values as measured by FACS of MCF7 +/− E2 +/− 10 μM PARPi. Red line = median. Statistical significance was determined by ordinary one-way ANOVA, followed by Sidak test. **g)** Relative change in median gH2AX intensities upon PARPi, shown as mean +/− s.d., n = 3 independent biological replicates. Statistical significance was determined by two-tailed unpaired t test. **h, i)** EM analysis of ssDNA gaps within replicating DNA of MCF7 +/− E2 +/− in vitro RNase H digestion. ssDNA gaps from 100 RI were quantified for size and distance to the fork junction. Thresholds: 30nt for size, 3 kb for distance. Dots in red ('pathological ssDNA gaps') correspond to gaps larger than 30nt and/or >3 kb distance to the fork. h) and i) display two independent biological replicates. **j)** EdU Alkaline Comet assay. Top: assay set up. Bottom: nascent strand (light grey) and genome-wide discontinuities (dark grey) in unperturbed conditions. Red lines: median olive tail moment. **k-m)** Half-lives of nascent strand discontinuities, derived from exponential fitting of data shown in Fig. 3g–i and shown as mean +/− s.d., n = 3 independent biological replicates. Significance determined by unpaired two-tailed t-test.

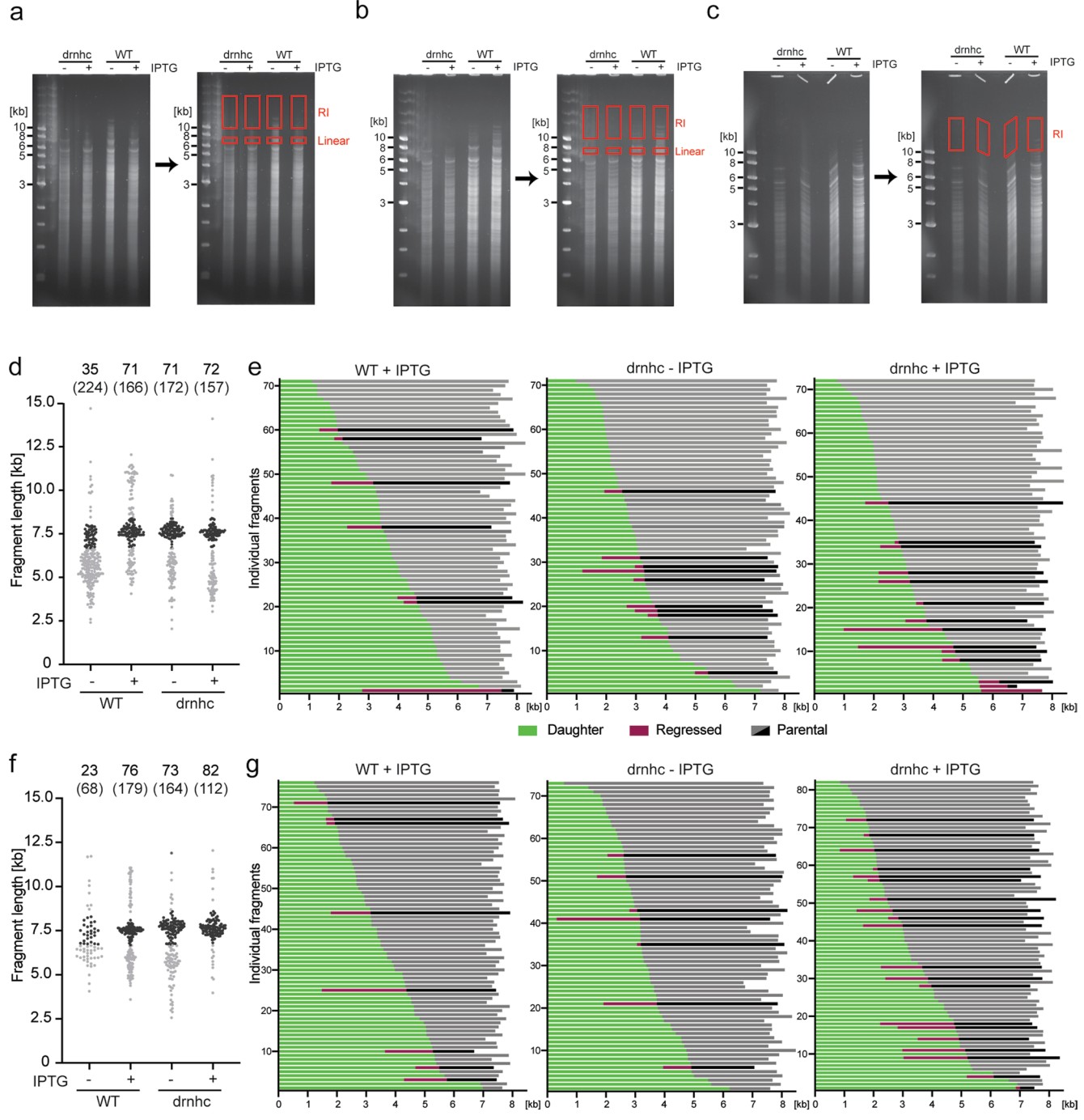

**Extended Data Fig. 5 | Replication forks stall and reverse while facing a transcription-replication conflict (TRC) in *Bacillus subtilis* (additional biological replicates). a)** EtBr-stained agarose gel of size separated genomic DNA from WT and *Δrnhc* mutant *B. subtilis* strains +/− IPTG. Fractions indicated in red were excised for further EM analysis (depicted in Fig. 4d, e)[49]. **b, d-e)** and **c, f-g)** represent two additional biological replicates of the replicate shown in Fig. 4. **b, c)** EtBr-stained agarose gel of size separated genomic DNA from WT and *Δrnhc* mutant *B. subtilis* strains +/− IPTG. Fractions indicated in red were excised for

further EM analysis. **d, f)** Fragment lengths of all imaged RI. The two numbers on top indicate the number of RI within the expected size range (dark dots) and the number of total RI imaged, respectively. **e, g)** Alignment of selected RI according to daughter strand length. Daughter strands are indicated in green, parental strands in grey. Reversed forks are labeled in pink/black, with pink marking length of the regressed arm and black the length of the parental strand prior to reversal.

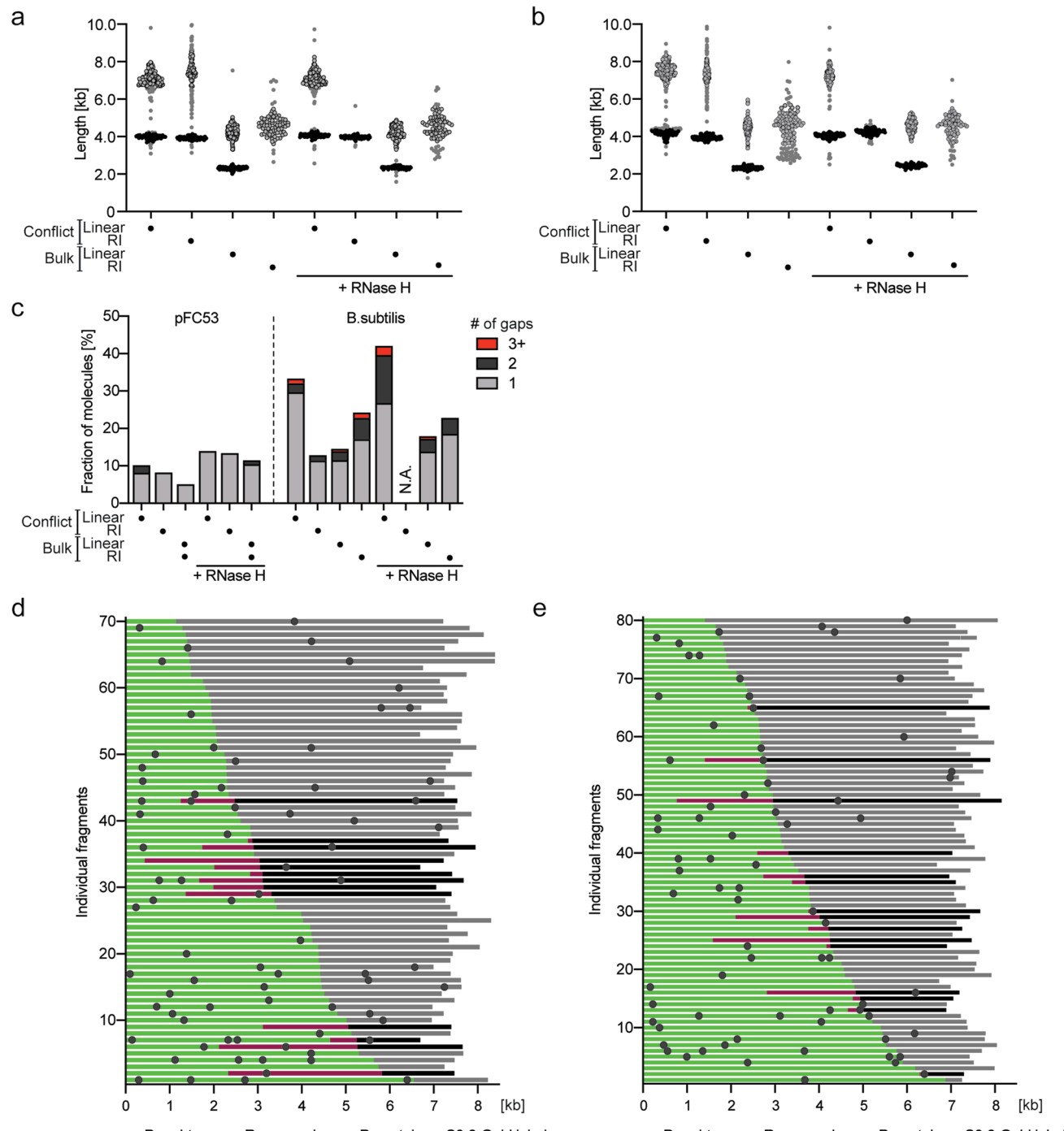

**Extended Data Fig. 6 | DNA:RNA hybrids accumulate within replicating conflict DNA in *Bacillus subtilis* (additional biological replicates). a, d)** and **b,e)** are independent biological replicates of Fig. 5 b/f. **a-b)** Length distribution of the imaged fragments. Black: pFC53; grey: *B. subtilis* material. **c)** Frequency and numbers of ssDNA gaps detected in the analyzed molecules in a). **d-e)** S9.6-gold binding position within the replicating conflict of *B. subtilis*. Green: daughter strand. Grey/black: parental strand. Pink: regressed arm. Dark grey dots: S9.6-gold label.

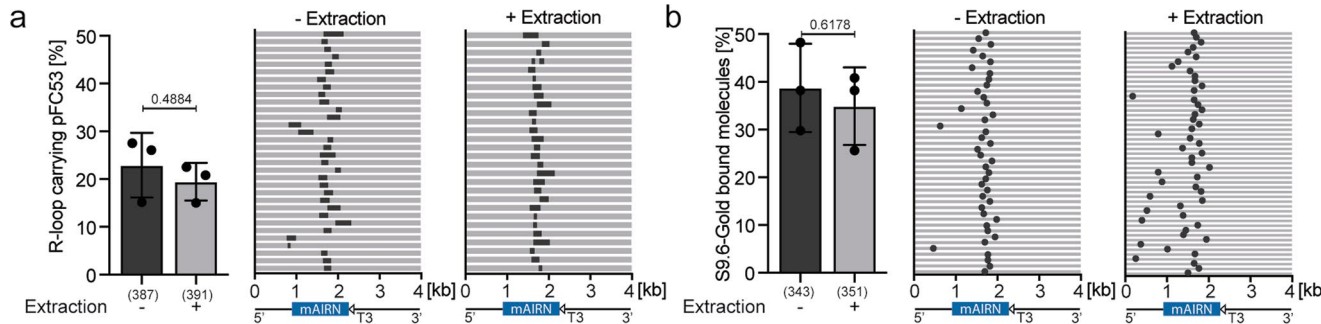

**Extended Data Fig. 7 | R-loops stability is maintained during gel-electrophorese and gel extraction. a)** Native EM analysis of linear pFC53 +/− gel extraction. Left: R-loop frequency, shown as mean +/− s.d., n = 3 independent biological replicates. Statistical significance determined by unpaired two-tailed t-test. Right: R-loop position. **b)** S9.6-gold immuno EM analysis of linear pFC53 +/− gel extraction. Left: S9.6-gold binding frequency, shown as mean +/− s.d., n = 3 independent biological replicates. Statistical significance determined by unpaired two-tailed t-test.

**a**

De novo transcription behind the fork

① 

Replisome bypass

②

RNA displacement and reannealing

③

**b**

Tethering of replicated duplexes to replisome

①

Hybrid processing and ssDNA-mediated fork reversal

②

Rad51 loading

ssDNA accumulation

Fork reversal

Incomplete chromatinisation, impaired fork progression

Nucleosome

③

**Extended Data Fig. 8 | See next page for caption.**

**Extended Data Fig. 8 | Possible mechanisms of post-replicative hybrid formation and hybrid-dependent replication interference. a) Possible mechanisms of post-replicative DNA:RNA hybrid formation**. Initiation of *de novo* transcription behind the replication fork generates nascent RNA, which can either anneal directly to exposed ssDNA on the lagging strand or reinvade the duplex daughter strand to form an R-loop, prone to nucleolytic processing (1). Alternatively, the RNA can be inherited from preexisting hybrids ahead of the fork either through replisome bypass (2) or through hybrid disassembly and subsequent RNA reannealing to the exposed ssDNA behind the fork (3). **b)**

**Possible mechanisms of replication interference by long/accumulating post replicative DNA:RNA hybrids**. DNA:RNA hybrids forming on the lagging strand may remain tethered to the replisome through RNA-binding and/or -processing enzymes, creating torsional constrains through DNA looping (1). Alternatively, hybrid processing may expose ssDNA, which induces RAD51 loading and subsequent fork reversal (2). Finally, excessive RNA:DNA hybrid formation may interfere with chromatinization of the newly synthesized daughter strand (3). In both a and b, the alternative mechanisms may coexist, contributing to the observed structures.

# Reporting Summary

## Statistics

For all statistical analyses, confirm that the following items are present in the figure legend, table legend, main text, or Methods section.

| n/a | Confirmed | |
|---|---|---|
| ☐ | ☒ | The exact sample size ($n$) for each experimental group/condition, given as a discrete number and unit of measurement |
| ☐ | ☒ | A statement on whether measurements were taken from distinct samples or whether the same sample was measured repeatedly |
| ☐ | ☒ | The statistical test(s) used AND whether they are one- or two-sided<br>*Only common tests should be described solely by name; describe more complex techniques in the Methods section.* |
| ☒ | ☐ | A description of all covariates tested |
| ☐ | ☒ | A description of any assumptions or corrections, such as tests of normality and adjustment for multiple comparisons |
| ☐ | ☒ | A full description of the statistical parameters including central tendency (e.g. means) or other basic estimates (e.g. regression coefficient) AND variation (e.g. standard deviation) or associated estimates of uncertainty (e.g. confidence intervals) |
| ☐ | ☒ | For null hypothesis testing, the test statistic (e.g. $F$, $t$, $r$) with confidence intervals, effect sizes, degrees of freedom and $P$ value noted<br>*Give P values as exact values whenever suitable.* |
| ☒ | ☐ | For Bayesian analysis, information on the choice of priors and Markov chain Monte Carlo settings |
| ☒ | ☐ | For hierarchical and complex designs, identification of the appropriate level for tests and full reporting of outcomes |
| ☒ | ☐ | Estimates of effect sizes (e.g. Cohen's $d$, Pearson's $r$), indicating how they were calculated |

*Our web collection on statistics for biologists contains articles on many of the points above.*

## Software and code

Policy information about availability of computer code

| | |
|---|---|
| Data collection | Dot blot/Immunoblot: Fusion-Capt Advance Solo7 (Vilber Lourmat); Electron microscopy: DigitalMicrograph Version 1.83.842 (Gatan, Inc.) or MAPS Version 3.16 (Thermo Fisher Scientific); DNA fiber imaging: Leica Application Suite X 3.6.0.20104; Flow cytometry: Attune NXT Software Version 4.2.0 |
| Data analysis | Dot blot, DNA fibers and Electron Microscopy (EM): ImageJ Version 2.0.0-rc-43/1.51h; Automated EM: MAPS Viewer Version 3.16; fork stitcher (https://github.com/jluethi/ForkStitcher, Version 0.1.1) and DNA content quantification algorithm (https://github.com/roessler-f/DNAQuantification); Flow cytometry: FlowJo Version 10.4; Olive tail moment: Open Comet plugin (version 1.3.1) for ImageJ; statistical analysis: Prism Version 8.4.2. |

For manuscripts utilizing custom algorithms or software that are central to the research but not yet described in published literature, software must be made available to editors and reviewers. We strongly encourage code deposition in a community repository (e.g. GitHub). See the Nature Portfolio guidelines for submitting code & software for further information.

## Data

Policy information about availability of data

All manuscripts must include a data availability statement. This statement should provide the following information, where applicable:
- Accession codes, unique identifiers, or web links for publicly available datasets
- A description of any restrictions on data availability
- For clinical datasets or third party data, please ensure that the statement adheres to our policy

Source data underlying all figures are provided in the source data file. Uncropped, original microscopy images and blots underlying all figures are provided as PDFs. Remaining original microscopy image data sets exceed several terabytes and will be made available upon reasonable request. The code for fork stitching is available

at: https://github.com/jluethi/ForkStitcher, Version 0.1.1. The code for the DNA content quantification algorithm can be found at: https://github.com/roessler-f/DNAQuantification.

# Field-specific reporting

Please select the one below that is the best fit for your research. If you are not sure, read the appropriate sections before making your selection.

☒ Life sciences        ☐ Behavioural & social sciences        ☐ Ecological, evolutionary & environmental sciences

For a reference copy of the document with all sections, see nature.com/documents/nr-reporting-summary-flat.pdf

# Life sciences study design

All studies must disclose on these points even when the disclosure is negative.

| | |
|---|---|
| Sample size | Sample size for all experiments shown (electron microscopy, n>70 in 2 or more independent experiments; DNA fibers, n>100 in 2 or more independent experiments) was chosen to obtain statistical power, in conformity to accepted standard sample size in a number of previous publications using these approaches:<br><br>Mijic et al., Nat Commun., DOI: 10.1038/s41467-017-01164-5<br>Vujanovic et al., Mol Cell, DOI: 10.1016/j.molcel.2017.08.010<br>Mutreja et al., Cell Rep., DOI: 10.1016/j.celrep.2018.08.019 |
| Data exclusions | N/A |
| Replication | For all experiments, the number of biological replicates is indicated in the figure legends and, without any exceptions, representative data shown in the figures was reproduced at least once. |
| Randomization | N/A |
| Blinding | Investigators were blinded for the data analysis of individual DNA fiber and Electron microscopy experiments. Blinding of the experimenter during data acquisition was not needed as most pipelines are automated, hence intrinsically unbiased. |

# Reporting for specific materials, systems and methods

We require information from authors about some types of materials, experimental systems and methods used in many studies. Here, indicate whether each material, system or method listed is relevant to your study. If you are not sure if a list item applies to your research, read the appropriate section before selecting a response.

## Materials & experimental systems

| n/a | Involved in the study |
|---|---|
| ☐ | ☒ Antibodies |
| ☐ | ☒ Eukaryotic cell lines |
| ☒ | ☐ Palaeontology and archaeology |
| ☒ | ☐ Animals and other organisms |
| ☒ | ☐ Human research participants |
| ☒ | ☐ Clinical data |
| ☒ | ☐ Dual use research of concern |

## Methods

| n/a | Involved in the study |
|---|---|
| ☒ | ☐ ChIP-seq |
| ☒ | ☐ Flow cytometry |
| ☒ | ☐ MRI-based neuroimaging |

## Antibodies

| | |
|---|---|
| Antibodies used | Primary antibodies:<br>Mouse S9.6 DNA:RNA hybrid, Kerafast, Cat # ENH001; AB_2687463;<br>Mouse S9.6-Gold NPS conjugate, BSI, this paper;<br>Mouse anti-dsDNA [HYB331-01], Abcam, Cat # ab27156; AB_470907;<br>Rat anti-BrdU (CldU) [BU1/75 (ICR1)], Abcam, Cat # ab6326; AB_305426;<br>Mouse anti-BrdU (IdU), clone B44 , Becton Dickinson, Cat # 347580; AB_10015219;<br>Mouse anti-gH2AX (Ser139), JBW301, Millipore, Cat # 05-636; AB_309864;<br>Rabbit anti-ZRANB3, Proteintech, Cat # 23111-1-AP; AB_2744527;<br>Mouse anti-actin, Sigma-Aldrich, Cat # A5441; AB_476744.<br><br>Secondary antibodies:<br>Goat anti-mouse-AlexaFluor488, Thermo Fisher Scientific, Cat # A-11001; AB_2534069;<br>Goat anti-mouse-AlexaFluor 647, Thermo Fisher Scientific, Cat # A-21235; AB_2535804;<br>Donkey anti-rat-Cy3, LubioScience, Cat # 712-166-153; AB_2340669; |

Anti-rabbit-HRP linked, VWR, Cat # NA934; AB_772206;
Anti-mouse-HRP linked, VWR, Cat # NA931; AB_772210.

Validation

The Mouse S9.6 DNA:RNA hybrid antibody has been tested in a plethora of studies for the use in affinity binding assays, ChIP, ChIP-seq, IP, Immunocytochemistry as claimed by the provider: s://www.kerafast.com/productgroup/432/anti-dna-rna-hybrid-s96-antibody. We validated this antibody in our own hands with RNase H controls.

The S9.6-Gold conjugate was validated in this study by electron microscopy, using an in vitro assay in combination with RNase H controls (Figure 1 and Extended Data Figure 1).

For the mouse anti-dsDNA antibody, the provider (Abcam, https://www.abcam.com/ds-dna-antibody-35i9-dna-bsa-and-azide-free-ab27156.html) has validated the following specificity: Primarily Double stranded DNA. Measurements by immuno-CE yielded KD's of 0.71 μM and 0.09 μM, for the interaction of this antibody with ss- and dsDNA, respectively. Strong reactivity with both ss- and dsDNA has been observed on dotblots as well as very weak reactivity with RNA.

The rat anti-BrdU antibody was validated for ICC/IF, IHC-P, Flow Cyt as reported by Abcam (https://www.abcam.com/brdu-antibody-bu175-icr1-proliferation-marker-ab6326.html) and has been in use and published by our own lab for the last decade (e.g. Mijic et al., Nat Commun., DOI: 10.1038/s41467-017-01164-5; Vujanovic et al., Mol Cell, DOI: 10.1016/j.molcel.2017.08.010; Mutreja et al., Cell Rep., DOI: 10.1016/j.celrep.2018.08.019).

The mouse anti-BrdU (IdU) antibody has been validated by the company (BD Biesciences, https://www.bdbiosciences.com/en-ch/products/reagents/flow-cytometry-reagents/clinical-discovery-research/single-color-antibodies-ruo-gmp/purified-mouse-anti-brdu.347580) and has been used and published by our own lab for the last decade (e.g. Mijic et al., Nat Commun., DOI: 10.1038/s41467-017-01164-5; Vujanovic et al., Mol Cell, DOI: 10.1016/j.molcel.2017.08.010; Mutreja et al., Cell Rep., DOI: 10.1016/j.celrep.2018.08.019).

The mouse anti-gH2AX has been tested in Western blotting, ICC, ChIP & Immunofluorescence (https://www.merckmillipore.com/CH/de/product/Anti-phospho-Histone-H2A.X-Ser139-Antibody-clone-JBW301,MM_NF-05-636-I?ReferrerURL=https%3A%2F%2Fwww.google.com%2F#).

The rabbit anti-ZRANB3 antibody has been KO and KD validated in many publications including several from our own lab: DOI: 10.1016/j.molcel.2017.08.010; DOI: 10.1038/s41467-017-01164-5; DOI: 10.1016/j.molcel.2019.10.026 and more as stated on the vendors website: https://www.ptglab.com/products/ZRANB3-Antibody-23111-1-AP.htm.

The monoclonal anti-beta-actin antibody has been claimed as validated for IF, IHC and protein arrays by Sigma-Aldrich: https://www.sigmaaldrich.com/CH/de/product/sigma/a5441.

All secondary antibodies were validated with a control missing the primary antibody.

# Eukaryotic cell lines

Policy information about cell lines

Cell line source(s)

Human MCF7 (kind gift from Karlene Cimprich, Stanford CA, USA); human HeLa TRIPZ control and human HeLa TRIPZ shTOP1 (gift from Philippe Pasero, IGH, France); U2OS cells (ATCC, HTB-96)

Authentication

None of the cell lines were authenticated in house for this manuscript.

Mycoplasma contamination

MCF7 and U2OS have been repeatedly tested negative for mycoplasma in our routine in-house tests.

Commonly misidentified lines
(See ICLAC register)

No commonly misidentified lines were used in this study.

