## [Peer Review File · Nature Structural & Molecular Biology]

Peer Review Information

Journal: Nature Structural & Molecular Information

Manuscript Title: Direct visualization of transcription-replication conflicts reveals post-replicative DNA:RNA hybrids

Corresponding author name(s): Massimo Lopes

Reviewer Comments & Decisions:

Decision Letter, initial version:

Message: 1st Oct 2021

Dear Massimo,

Thank you for submitting your manuscript "Direct visualization of transcription-replication conflicts reveals post-replicative DNA:RNA hybrids". I apologize for the delay while we awaited the reports (copied below) of the 3 reviewers, all experts in DNA replication and genome instability, who have evaluated your manuscript. Unfortunately, after carefully considering their comments, we cannot offer to publish your manuscript, at least in its current form, in Nature Structural & Molecular Biology.

You will see that while all 3 referees appreciate the direct visualization of R loops in genomic DNA, they also find that additional experimentation and controls are needed to resolve some apparent discrepancies in the data and to convincingly demonstrate that estrogen stimulation reproducibly induces a transcriptional burst in human cells. Reviewers 2 and 3 are positive about the EM approach and potential interest of the findings, and they offer specific suggestions to strengthen the central conclusions of the work. Reviewer 1 queries aspects of both the methodology and data interpretation that we agree should be clarified in a potential revised manuscript.

If further experimentation, analysis, and revisions would allow you to address the referees concerns in full, we would be prepared to consider an appeal of our decision, on the condition that no related work is published in the interim or has been accepted in our journal. Please don't hesitate to contact me to discuss an appeal and potential revision. Please note that, until we have the opportunity to read the revised manuscript in its entirety, we cannot promise that it will be sent back for peer review.

I am sorry we could not be more positive on this occasion. I hope that you find the referees' comments useful in deciding how best to proceed.

With kind regards,

Beth

Beth Moorefield, Ph.D.
Senior Editor
Nature Structural & Molecular Biology

Reviewers' Comments:

Reviewer #1:

Remarks to the Author:

Stoy et al use an EM platform to visualize outcomes to replication through transcription-replication conflicts [TRC(s)] in a human cell line as well as at an inducible head-on collision between a replisome and a transcription complex in *B. subtilis*. They develop a protocol that retains R-loop/RNA:DNA hybrid integrity during sample preparation and analyze replication intermediates for the presence and location of hybrids by native and immune-EM. They show that the hybrids are not specifically found to be actively blocking replication forks. Rather, the hybrid signals are found behind forks, suggesting that the RNA-DNA hybrids are processed post-replicatively.

1. The authors have demonstrated that they can visualize R-loops in genomic DNA both directly and by use of an antibody to RNA-DNA hybrids, although the latter method is clearly far less reliable than the former. Unfortunately, the evidence offered that a transcriptional burst increases the extent of R-loops in the region being transcribed is simply not reproducible. The amount of S9.6 IP normalized to a negative locus shown in Fig. 2a differs by a factor of almost 10 with the same quantitation of the biological replicate shown in Extended Data Fig. 3a. Furthermore, in Fig. 2A, there does not appear to be any difference between the effect of added estrogen between inducible and constitutive loci. Whereas it is the case that both replicates show R-loops, the lack of reproducibility of the 'platform' raises questions with respect to what it actually delivers.

2. It is also not at all clear how the values for R-looped DNA molecules determined by native EM were arrived at. P. 30, l. 776 indicates that "Samples were annotated for molecules of interest using" What does that mean? Were molecules selected because they looked like they carried an R-loop? If so, that is not an unbiased approach. Similarly, referring to the immune-EM, p. 31, l. 795 indicates that "for each experimental condition, at least 70 molecules were analyzed." What does that mean with respect to how they were selected? Again, this does not sound like an unbiased approach.

3. The effect of the transcriptional burst on replication fork progression, which the authors ascribe to the formation of R-loops that may, in turn, lead to replication fork reversal, is quite remarkable. The authors detail that the number of R-loops per cell increases from 614 to 830, on average, with the transcriptional burst. Based on Fig. 3a, this change in R-loop burden of about 35% leads to a decisive, statistically significant, reduction in the overall rate of replication fork progression in a cell where 30,000 – 50,000 origins have to fire to replicate the genome. How do the authors think that happens, exactly?

4. Extended data Fig. 1e shows that more than half of the S9.6 ab binding is not R-loop/RNA-DNA hybrid specific under dilute concentrations of R-loops, a condition that prevails in MCF7 cell lines. So, what can one say about the values for R-loop density? The authors appear to be ignoring one of their own caveats.

5. Lang et al, (ref. 9) Figs. 2E and F showed that there was under-replication in an area of up to 20-30 kbp around the TRC, suggesting that stalling may occur in a wider area than the 7.5 kbp selected by the authors for EM. So, instead of finding only hybrids that were bypassed by the fork and not any direct fork collisions, maybe they just didn't look far enough. Two of the authors of Lang et al. (ref. 9) are also authors of the manuscript under review, it seems to me that the current efforts should match the previous data.

6. The authors argue that whereas R-loops do form in front of the replication fork, they tend to be 'processed' such that only the RNA-DNA hybrid is left behind, which the replication fork is able to bypass, albeit with an overall slowing of fork progression that the authors attribute to fork reversal. The authors, however, do not show any RI's from the *B. subtilis* TCR in which a reversed fork could be visualized opposed to a head-on R loop. One presumes that the reason for this is related to their failure to recover any gapped DNA from the RNase H-treated RI fraction. The authors hypothesize that "RNase H digestion liberated massive stretches of ssDNA on *B. subtilis* RIs, thereby compromising their integrity and recovery for EM analysis." This statement implies that the R-loops that were present were equally "massive." If so, surely the authors could devise some other EM approach that would reveal their existence, a prep step that didn't involve a kit, perhaps. All this gives me cause to wonder what the "processing" steps are in vivo that the authors invoke? One would have assumed if the R-loops were "massive" and had extensive stretches of ssDNA, that ssDNA would have been coated with SSB, affording some protection. Furthermore, the data displayed in Fig. 5d shows that less than 50% of the molecules cartooned with fork reversal have any S9.6 antibody bound. The authors main conclusion therefore appears to be based on weak evidence.

Reviewer #2:

Remarks to the Author:

This is a nice and worth publishing paper from a collaborative effort of the Lopes' lab. The main and relevant result is that authors are able to visualize by EM DNA-RNA hybrids formed in vivo in bacteria and human cells, apart from others formed in vitro in a plasmid in which transcription is induced by a T3 promoter. To be able to detect the hybrids, authors use the S9.6 monoclonal antibody and conditions known to increase the levels of hybrids in HeLa and *Bacillus* cells at specific regions where it was previously reported an accumulation of hybrids. The manuscript is an important contribution to the field with the collateral message, that the S9.6 antibody is indeed an excellent tool to detect hybrids when used properly. This is an aspect that I would suggest the authors to remark in their conclusions also.

Besides this core conclusions that justify already the publication, the authors make a nice a thorough analysis of the presence of hybrids in specific regions with particular emphasis at regions in which transcription-replication conflicts occur, leading to a number of possible models that are discussed. I have several suggestions below that I believe should contribute to enhance the strength of the conclusions and to clear up few aspects that may weaken some conclusions for not considering other options or trying to go too far.

1. Fig. 2 and related text. While the impact of estrogen in R loop accumulation was reported previously, and certainly, authors provide repeated experiments of EM, it is unclear why DRIP data (2a) are not plotted putting together 3 experiments with mean and SD. I am concerned about the difference in values between the two experiments shown, for which E2 addition causes an almost 10x higher increase in Extended Fig. 3 compared to Fig. 2a. This is particularly important, because Fig. 2e is made by compiling the data of the 2 independent replicates. With this in mind, as presented, authors should provide the plot of Fig. 2e for each experiment independently in Extended Fig. 2, so that the reader can see whether lengths and frequencies are the same in both experiments.
2. Fig. 2d. Why the scale bar here is shown in bp instead of nm like the others. I guess this is a typo.
3. Fig. 2e. It would be more informative to indicate (at least in addition) the median value to know what is the most common size. This will be significantly smaller than 200 bp, as has been shown in other organisms. The way the text is written suggests that hybrids are generally very long.
4. Fig. 3a-c. Authors show that E2 addition increases DNA breaks and affects replication, consistent with the idea that E2 stimulates hybrids. However, authors should include the experiment with RNH to show that indeed this is linked to hybrids and not just the burst of active RNA polymerases in the DNA. Otherwise, it is not clear what these results add that we do not know already.
5. Fig. 3d-e. The previous point is important because the length of gaps seems much shorter than the hybrids measured in Fig. 2e. Authors should discuss this fact. Could it be possible that gaps detected are due to unprocessed RNA primers of Okazaki fragments, especially for the very short gaps. The data shows a distribution, so that an increase in size due to RNH digestion will enlarge short gaps that would be moved to the large size section. The way to solve this issue is providing the control data of E2 non-treated samples. I think this is important, because the pattern should change dramatically if those gaps reflect hybrids resulting from the burst of transcription after E2 treatment.
6. Fig. 4d-e. The control -IPTG is required. The increase of reversed forks in drnhc bacteria is meaningful if not observed in the wt. Otherwise, conclusions are not solid.
7. Fig. 4d-e. It is unclear from the data the increase observed in the plot of reversed forks in 4e in drnhc cells, since the data in 4d shows 11/70 versus 14/70 reversed forks containing fibers. I assume I am missing something. Explain, please.
8. Fig. 5d-f. Here is where the problem of Okazaki fragments becomes more important. The only way to detect the hybrids in those experiments was by S9.6 labeling. Since the displaced ssDNA seems not to be really detected, the assumption would be that hybrids are very short, in contrast with the data in vitro and from HeLa cells (Figs. 1 and 2). This together with the overabundance of gaps in the daughter DNAs makes likely that a good portion of such signals correspond to Okazaki primers rather than newly generated hybrids. Thus, it is essential to show at least the analysis of -IPTG samples to check whether indeed such daughter S9.6 signal overabundance is observed.
9. Fig. 5e legend. It refers to Extended Data Fig. 7 when it should be 5.

Discussion

1. Major differences between promoting hybrid formation via estrogen, in which a burst of transcription would cause hybrids that would not form that easily in wt cells, and hybrids detected in *rnhC* mutants (certainly under transcription conditions established by IPTG) that reveals hybrids formed normally in WT cells that are not processed due to the lack of RNH. Thus, authors are putting together two set of results that may not be related and indeed show differences, so that conclusions may not be unique to all. This needs to be taken into account and discussed properly.

2. Authors should discuss possible differences between bacteria and HeLa cells and between hybrids behind the replication fork and the rest. With the issue of Okazaki's the conclusion that hybrids are formed or accumulated more often is not convincing. Please, keep in mind that hybrid length has been established with the bisulfite mutagenesis by different authors in the Ig genes, yeast or HeLa cells. This technique relies on the sequencing of the mutagenized displaced ssDNA, which means they detect R loops. The study in ref. 46 shows a nice correlation between S9.6 maps and bisulfite maps, abounding in the idea that most hybrids seem to be in the form of R loops. This is consistent with in the *in vitro* and HeLa +E2 system is where clearly R loops can be visualized, but contrasts with the conclusions here referring to hybrids as the most prominent event and contributing to the uncertainty about the nature of the assymmetric distribution of S9.6 signals around forks, which could be partially explained by Okazaki primers. This needs to be considered and discussed, and the control experiments suggested should help clarify.

3. Authors affirms that transcription is silenced behind the fork. However, this has been defined in eukaryotes in connection to chromatin structure (ref. 48). How would this silencing be achieved in *Bacillus*? Provide a reference or clarify this issue.

4. This is an impressive amount of great work, but it would be nice in the future to study the co-directional system to probe part of the models. Since it is well known that the impact of head-on is much higher than the co-directional, it is not easy to understand why the codirectional hybrids, accumulated at the future leading strand will not suppose a block to the progression of the fork, while head-on will do if most of hybrids are found behind the fork, from which they are supposed to delay replication. The RNA polymerase is the first physical obstacle that the fork encounters in head-on. Could it be that the pause of the fork itself may contribute to the formation of the short hybrids behind the fork as a consequence of unprocessed Okazaki primers?. It would be nice if authors can add some light with their Discussion on this fact.

5. Along with their line of thinking, authors should include in their Discussion the recent paper showing indeed that hybrids may have different origins and processed by different factors PMID: 34294712

6. It has recently been published (PMID: 33657379) that topological stress generated at head-on collisions causes R loops ahead of the fork, events that are controlled by type II topoisomerases in *Bacillus*. This new study, however, shows in the same bacteria sp, that hybrids seems to be poorly seen ahead of the fork. Any hints? A discussion would be nice, because it is not clear at all how a hybrid behind the fork should delay fork movement, as

its connection with reversed forks is not too clear, since most S9.6 signals (Fig. 5 and Extended Fig.5 shows most S9.6 signal not associated with fork reversal)

7. ref 27 should be cited when referring to model in Extended Fig. 7a 1. In that reference it was not proposed that the fork is able to pass the hybrids, as the authors indicate in Intro and Discussion (please correct), but that indeed hybrids are formed behind the forks in the lagging strand with an RNA molecule produced in the leading strand and if so, such putative hybrids behind the fork not affecting fork movement. This needs to be properly referred.

8. One interesting proposal in this study is that hybrids seen ahead of the fork may be the product of a previous R loop in which the displaced ssDNA is removed. Do authors think that NER endonucleases would do this as has been shown (ref. 18). Has anybody studied TRCs in UvrC mutants in Bacillus? Wpuld result fit with that notion?

In summary, excellent work. I hope my comments contribute to make it stronger.

Reviewer #3:

Remarks to the Author:

This is a nicely written manuscript describing a novel electron microscopy (EM)-based approach to directly visualize R-loops formed during head-on transcription replication conflicts. The authors start by showing that R-loops can be visualized in vitro by EM and that these structures are characterized by a ssDNA strand, which is displaced from the RNA:DNA duplex. Conversely, RNA:DNA hybrid filaments cannot be distinguished from the surrounding DNA by EM. However, they show that RNA:DNA hybrids can be detected by EM using an S9.6-gold labeling approach. The same EM-based approach was then successfully applied to visualize R-loops in a mammalian breast cancer cell line (MCF7), where estrogen stimulation causes a transcriptional burst that leads to R-loop accumulation. From this analysis, the authors conclude that larger R-loops, longer than 300bp, are more frequently induced by the transcriptional burst compared to smaller R-loops, although the molecular basis for this difference remain unclear. Using this new approach, the authors estimate the total R-loop burden on the genome caused by the same transcriptional burst. Next, they investigate the effect of aberrant R-loop accumulation on DNA replication. In agreement with previous findings, they observe that the transcriptional burst perturbs replication fork progression and suggest that this fork slowing is associated with reversed fork formation because the fork slowing phenotype is rescued by PARP inhibition. By inspecting the structure of the replication intermediates by EM, they also observe increased accumulation of ssDNA gaps behind the replication forks upon treatment with RNase H, suggesting that the transcriptional burst leads to hybrid accumulation behind the forks. Finally, they expand their analysis to study how RNA:DNA hybrids interfere with DNA replication in a bacterial model system. From this analysis, they confirm that replication forks frequently reverse when facing transcription replication conflicts and that RNA:DNA hybrids accumulate behind the replication forks, suggesting that these structures are efficiently bypassed by the replisome. Overall, this an important study that provides the first direct visualization of R-loop structures in cells and new insights into the mechanisms by which replication forks deal with these intermediates. I think that these findings will spark the interest of many groups studying transcription replication collisions and will open new avenues of research in the field. However, the

authors need to include additional experiments and missing controls to strengthen their conclusions and increase the impact of their findings.

Major criticisms:

1. The extended data of Figure 1E shows almost 50% nonspecific S9.6 binding when using 50% of the competitor the plasmid that lacks R-loops. This result undermines the interpretation of the following results obtained with the S9.6 antibody, particularly the results of Figure 5d where the authors map the location of RNA:DNA hybrids on replication forks using the S9.6 label.

2. Figure 2. The authors should discuss the possibility that longer R-loops are more reproducibly increased compared to shorter R-loops simply because they might be more stable and more likely to persist during the DNA extraction process compared to shorter R-loops.

3. The two replicates of the experiments showing the frequency of small R-loops (Extended Data Figure 3e) show very different results. In particular, the first repeat does not show any significant increase in R-loops after estrogen stimulation. These experiments should be repeated in triplicate to confirm that the results are reproducible.

4. Figure 3. The authors should test whether the fork slowing phenotype associated with estrogen induction is rescued by addition of RNase H to provide further support to their model that this phenotype is linked to accumulation of RNA:DNA hybrids.

5. Figure 3. The authors' model that RNase H treatment leads to accumulation of ssDNA gaps behind the replication forks should be strengthened by additional DNA fiber assays in the presence of the S1 nuclease, which would cause tract shortening if ssDNA gaps are present on the thymidine labeled tracts.

6. Figure 3a. PARP has several cellular functions, in addition to the previously reported role in reversed fork stabilization discussed in the manuscript. The authors should repeat the DNA fiber experiments of Figure 3a by depleting the SMARCAL1 translocase to prevent formation of reversed forks and provide further support to their model that fork slowing is indeed associated with reversed fork accumulation. The same experiments could be extended to other translocases involved in fork reversal such as ZRANB3 or HLF. Moreover, the authors should repeat the same assays by depleting the RECQ1 helicase, which should prevent reversed fork stabilization.

7. Figure 3a. The authors should measure the length of IdU tracts and confirm that the transcriptional burst leads to the same decrease in tract length observed when measuring the CldU tracts.

8. Figure 5d shows S9.6 binding to the parental strand and one of the two daughter strands of different replication intermediates. An additional figure showing the same results for the complementary daughter strand would be informative to help the reader understanding whether S9.6 binding can simultaneously occur on both strands.

9. For the experiments in *Bacillus subtilis*, the authors state that "the delta-rnhc mutant strain lacks endogenous RNase H activity". However, the authors should clarify that this strain only lacks RNase H III activity, and still retains some residual RNase H activity

mediated by RNase HII.

10. Some of the figures are missing essential controls. In particular, Figure 3d should include the untreated control (-E2). The same applies to Figure 4d that lacks the (WT-IPTG) control.

Minor comments:

1. Figure 2d is not referenced in the text.

Although we cannot publish your paper, it may be appropriate for another journal in the Nature Portfolio. If you wish to explore the journals and transfer your manuscript please use our [Redacted] manuscript transfer portal. If you transfer to Nature journals or the Communications journals, you will not have to re-supply manuscript metadata and files. This link can only be used once and remains active until used.

All Nature Portfolio journals are editorially independent, and the decision on your manuscript will be taken by their editors. For more information, please see our <Redacted>manuscript transfer FAQ page.

Author Rebuttal to Initial comments

Reviewers' Comments:

Reviewer #1:

Remarks to the Author:

Stoy et al use an EM platform to visualize outcomes to replication through transcription-replication conflicts [TRC(s)] in a human cell line as well as at an inducible head-on collision between a replisome and a transcription complex in *B. subtilis*. They develop a protocol that retains R-loop/RNA:DNA hybrid integrity during sample preparation and analyze replication intermediates for the presence and location of hybrids by native and immuneEM. They show that the hybrids are not specifically found to be actively blocking replication forks. Rather, the hybrid signals are found behind forks, suggesting that the RNA-DNA hybrids are processed post-replicatively.

1. The authors have demonstrated that they can visualize R-loops in genomic DNA both directly and by use of an antibody to RNA-DNA hybrids, although the latter method is clearly far less reliable than the former.

We have now clarified within both Results and Discussion the limitations of the S9.6 antibody for immuno EM. In conditions of sufficient DNA:RNA hybrid enrichment, this proved as a precious tool to directly demonstrate post-replicative DNA:RNA hybrids, and may be used by other researchers in similar experimental conditions. We have published in the meanwhile a book chapter (Stoy et al., Methods in Molecular Biology 2022a), referenced in the revised manuscript, that describes in detail this method, and its potential and limitations.

Unfortunately, the evidence offered that a transcriptional burst increases the extent of R-loops in the region being transcribed is simply not reproducible. The amount of S9.6 IP normalized to a negative locus shown in Fig. 2a differs by a factor of almost 10 with the same quantitation of the biological replicate shown in Extended Data Fig. 3a. Furthermore, in Fig. 2A, there does not appear to be any difference between the effect of added estrogen between inducible and constitutive loci. Whereas it is the case that both replicates show R-loops, the lack of reproducibility of the 'platform' raises questions with respect to what it actually delivers.

The variability in absolute values for DRIP experiments is well known in the field, although the trends we have discussed in the manuscript are clearly reproducible. The particular variability pointed out by the referee relates to two replicates of DRIP experiments performed on the same material used for EM analysis, which were performed quite some time apart. We have now repeated in triplicate the same experiment and included it in Fig. 2a, showing that biological repetitions performed within a short time period and with the same batch of reagents do yield highly reproducible results. For completeness, we are still showing the individual DRIP experiments performed in parallel to the EM analyses, in Extended Data Fig. 3a/b. DNA:RNA hybrid induction by E2 also in constitutive loci was previously shown by Stork et al. and reflects the widespread induction of transcription by E2, going beyond the specific E2-inducible genes we have tested. Hybrid induction at constitutive loci did not prove statistically significant in the three biological replicates performed in revision. The text has been changed accordingly, but this point is anyway not relevant for any of the key conclusions of our study.

2. It is also not at all clear how the values for R-looped DNA molecules determined by native EM were arrived at. P. 30, l. 776 indicates that "Samples were annotated for molecules of interest using" What does that mean? Were molecules selected because they looked like they carried an R-loop? If so, that is not an unbiased approach. Similarly, referring to the immune-EM, p. 31, l. 795 indicates that "for each experimental condition, at least 70 molecules were analyzed." What does that mean with respect to how they were selected? Again, this does not sound like an unbiased approach.

We have in the meanwhile published a book chapter (Stoy et al., *Methods in Molecular Biology* 2022b) – referenced in the revised manuscript - that describes in detail our EM pipeline for R-loop visualization, including criteria for R-loop identification and statistical considerations. By definition, EM methods are based on visual inspection and operator-based interpretation of EM images. However – as described in the book chapter – the entire DNA content of a large area of the EM grids is automatically imaged and carefully inspected, to select molecules that deserve in depth visualization and interpretations. Those points of interest are then typically analyzed in a blinded manner by several experimenters, in order to consolidate shared interpretations of the images. We are not aware of any methodology that can completely eliminate arbitrary interpretations of the images in EM analysis, as well as most other imaging-based methods.

3. The effect of the transcriptional burst on replication fork progression, which the authors ascribe to the formation of R-loops that may, in turn, lead to replication fork reversal, is quite remarkable. The authors detail that the number of R-loops per cell increases from 614 to 830, on average, with the transcriptional burst. Based on Fig. 3a, this change in R-loop burden of about 35% leads to a decisive, statistically significant, reduction in the overall rate of replication fork progression in a cell where 30,000 – 50,000 origins have to fire to replicate the genome. How do the authors think that happens, exactly?

We have now included a paragraph in the final section of the Discussion, referring to previous work from our lab, showing that reversal can rapidly extend to many more forks than those directly challenged by the source of stress (in this case DNA:RNA hybrids). Although the molecular mechanisms mediating this global nuclear response are still being clarified, we consider this as the most likely explanation for the reversal-dependent slowing we report in these conditions.

4. Extended data Fig. 1e shows that more than half of the S9.6 ab binding is not R-loop/RNA-DNA hybrid specific under dilute concentrations of R-loops, a condition that prevails in MCF7 cell lines. So, what can one say about the values for R-loop density? The authors appear to be ignoring one of their own caveats.

As mentioned above, we highlighted and discussed the limitations with the use and specificity of the S9.6 antibody, which are also by now well known in the field. However, as we now clarified in the main text and the relevant figure legends, all key conclusions of our study either stem from the use of S9.6 under controlled

conditions of specificity (Fig. 5, where internal controls prove specific S9.6-Gold specific binding, as in Extended Data Fig. 1d-f) or prescind from the use of S9.6 and are rather based on visual inspection of DNA:RNA molecules by EM or other S9.6-independent methods. This specifically applies to all data on MCF7, where S9.6-based methods were simply used as a control, to show that we can reproduce DNA:RNA hybrid accumulation as previously published, and we can maintain their stability across our complex EM procedures.

5. Lang et al, (ref. 9) Figs. 2E and F showed that there was under-replication in an area of up to 20-30 kbp around the TRC, suggesting that stalling may occur in a wider area than the 7.5 kbp selected by the authors for EM. So, instead of finding only hybrids that were bypassed by the fork and not any direct fork collisions, maybe they just didn't look far enough. Two of the authors of Lang et al. (ref. 9) are also authors of the manuscript under review, it seems to me that the current efforts should match the previous data.

We were forced for technical reasons to restrict our investigations to a reasonably small fragment, compatible with isolation of replication intermediates from the gel and with visual inspection of unfolded DNA molecules. We cannot formally exclude that specific events may occur beyond the fragment we focused on. However, we note that all molecular phenotypes described by Lang et al (copy number analysis, ChIP-seq, DRIP-seq) were clearly detectable also (and in fact primarily) within the conflict region we observed. Accordingly, we do detect clear signs of replication fork slowing and accumulation across this region and we can prove a drastic accumulation of post-replicative DNA:RNA hybrids at those replication intermediates. Yet, despite proven stability of the hybrids during sample preparation and their direct detection behind replication forks, we found no evidence for canonical R-loops accumulating ahead of forks. It is thus highly unlikely that we missed crucial alternative events taking place on other DNA fragments adjacent to the conflict region.

6. The authors argue that whereas R-loops do form in front of the replication fork, they tend to be 'processed' such that only the RNA-DNA hybrid is left behind, which the replication fork is able to bypass, albeit with an overall slowing of fork progression that the authors attribute to fork reversal. The authors, however, do not show any RIs from the *B. subtilis* TCR in which a reversed fork could be visualized opposed to a head-on R loop. One presumes that the reason for this is related to their failure to recover any gapped DNA from the RNase H treated RI fraction. The authors hypothesize that "RNase H digestion liberated massive stretches of ssDNA on *B. subtilis* RIs, thereby compromising their integrity and recovery for EM analysis." This statement implies that the R-loops that were present were equally "massive." If so, surely the authors could devise some other EM approach that would reveal their existence, a prep step that didn't involve a kit, perhaps. All this gives me cause to wonder what the "processing" steps are in vivo that the authors invoke? One would have assumed if the R-loops were "massive" and had extensive stretches of ssDNA, that ssDNA would have been coated with SSB, affording some protection. Furthermore, the data displayed in Fig. 5d shows that less than 50% of the molecules cartooned with fork reversal have any S9.6 antibody bound. The authors main conclusion therefore appears to be based on weak evidence.

Several lines of evidence in our paper demonstrate that our EM procedure does not "resolve" DNA:RNA hybrids or R-loops: 1) R-loops are promptly and abundantly detected on in vitro-transcribed plasmids, prone to form R-loops (Fig. 1); 2) None of the preparation steps for our EM analysis on genomic DNA leads to a decrease of S9.6 binding to DNA (Extended Data Fig. 2); 3) R-loops can be directly visualized by EM and their calculated frequency on the genome is overall in good agreement with recent publications (Fig. 2 and references in main text); 4) RNase H digestion prior to EM visualization induces detectable changes in replication intermediate architecture (ssDNA gaps; Fig. 3 and Extended Data Fig. 4); 5) hybrids are directly detected by immuno EM on *B. subtilis* replication intermediates, under controlled conditions of binding specificity. There is therefore no reason to imagine that a specific subset of structures (e.g. canonical R-loops ahead of replication forks) is specifically lost from our preps or undetectable in our EM analyses.

Admittedly, we would have much preferred to be able to retrieve *B. subtilis* replication intermediates from the conflict region after RNase H digestion, which may have directly shown drastic accumulation of ssDNA regions due to hybrid processing. Unfortunately, this proved impossible despite numerous attempts. However, based on all other data in Fig. 5 - primarily, the direct detection of abundant hybrids on replicated duplexes - we consider most likely that massive release of ssDNA leads to inter-molecular aggregation in vitro and prevents recovery of the material. Please note that our EM visualization method requires extensive deproteinization and in this context SSB proteins may not protect/shield the ssDNA generated in vitro. Moreover, as described in the methods (and

now also in the recently published book chapter, cited in the revised manuscript), the procedure does not use kits for DNA isolation, but controlled isolation of replication intermediates from different areas of agarose gels, which proved consistently compatible with EM visualization of the corresponding intermediates.

Finally, our general model (Fig. 6) – as well as any of its specific variants, included in Extended Data Fig. 8 – does not imply that reversal is required for R-loops processing/bypass, but rather that the bypass leads to post-replicative hybrids, which may eventually lead to ssDNA accumulation and topological constraints, finally promoting reversal. In this respect, and given the technical considerations discussed above, we think that lack of visualization of reversed forks proximal to canonical R-loops is neither a technical problem, nor necessarily surprising. We rather think that this “negative” evidence indirectly supports an alternative sequence of events, which is directly supported by several lines of “positive” evidence in our manuscript.

Reviewer #2:

Remarks to the Author:

This is a nice and worth publishing paper from a collaborative effort of the Lopes' lab. The main and relevant result is that authors are able to visualize by EM DNA-RNA hybrids formed in vivo in bacteria and human cells, apart from others formed in vitro in a plasmid in which transcription is induced by a T3 promoter. To be able to detect the hybrids, authors use the S9.6 monoclonal antibody and conditions known to increase the levels of hybrids in HeLa and Baccillus cells at specific regions where it was previously reported an accumulation of hybrids. The manuscript is an important contribution to the field with the collateral message, that the S9.6 antibody is indeed an excellent tool to detect hybrids when used properly. This is an aspect that I would suggest the authors to remark in their conclusions also.

We thank this reviewer for the positive evaluation of our work and its relevance. We agree with the importance of this final technical aspect, have included a sentence in discussion to stress the potential and reliability of the S9.6 antibody and provided internal controls for specificity.

Besides this core conclusions that justify already the publication, the authors make a nice a thorough analysis of the presence of hybrids in specific regions with particular emphasis at regions in which transcription-replication conflicts occur, leading to a number of possible models that are discussed. I have several suggestions below that I believe should contribute to enhance the strength of the conclusions and to clear up few aspects that may weaken some conclusions for not considering other options or trying to go too far.

1. Fig. 2 and related text. While the impact of estrogen in R loop accumulation was reported previously, and certainly, authors provide repeated experiments of EM, it is unclear why DRIP data (2a) are not plotted putting together 3 experiments with mean and SD. I am concerned about the difference in values between the two experiments shown, for which E2 addition causes and almost 10x higher increase in Extended Fig. 3 compared to Fig. 2a. This is particularly important, because fig. 2e is made compiling the data of the 2 independent replicates. With this in mind, as presented, authors should provide the plot a Fig. 2e for each experiment independently in Extended Fig. 2, so that the reader can see whether lengths and frequencies the same in both experiments.

The variability in absolute values for DRIP experiments is well known in the field, although the trends we have discussed in the manuscript are clearly reproducible. The particular variability pointed out by the referee relates to two replicates of DRIP experiments performed in parallel to the EM analysis, which were performed quite some time apart. We have now repeated in triplicate the same experiment and included it in Fig. 2a, showing that biological repetitions performed within a short time with the same batch of reagents do yield highly reproducible results. For completeness, as requested by this reviewer, we are still showing the individual DRIP experiments performed in parallel to the EM analyses, in Extended Data Fig. 3a/b.

2. Fig. 2d. Why the scale bar here is shown in bp instead of nm like the others. I guess this is a typo. As Fig. 2e (R-loop length) plots values in bp, we find important to include a scale bar with the same unit in Fig. 2d. However, for completeness, we are now indicating length of the scale bar in both nm and bp.

3. Fig. 2e. It would be more informative to indicate (at least in addition) the median value to know what is the most common size. This will be significantly smaller than 200 bp, as has been shown in other organisms. The way the text is written suggest that hybrids are generally very long.

We thank the reviewer for this important remark. We have included a representation of the median value in the figure and we adapted the text to stress that most R-loops are indeed smaller than 200bp.

4. Fig. 3a-c. Authors show that E2 addition increases DNA breaks and affects replication, consisting with the idea that E2 stimulates hybrids. However, authors should include the experiment with RNH to show that indeed this is linked to hybrids and not just the burst of active RNA polymerases in the DNA. Otherwise, it is not clear what these results add that we do not know already.

We have now performed the experiment suggested by this reviewer, transiently overexpressing RNase H1 in MCF7 cells. Importantly, even moderate expression of RNase H1 in replicating cells (Extended Data Fig. 4a)

leads to a marked rescue of fork speed in presence of E2, clearly linking DNA:RNA hybrid accumulation with delayed fork progression. RNase H1 had also a detectable effect on fork speed in absence of E2; this most likely reflects residual accumulation of E2-dependent hybrids despite E2-deprivation. As this point is relevant also for other aspects of this manuscript (see below), we now comment throughout the revised manuscript on the intrinsic limitations of this experimental system, where cells are essentially dependent on E2 for growth and experience clear signs of stress when E2 is deprived. As argued within the main text, we reckon that this intrinsic limitation does not affect any of our major conclusions.

5. Fig. 3d-e. The previous point is important because the length of gaps seems much shorter than the hybrids measured in Fig. 2e. Authors should discuss this fact. Could it be possible that gaps detected are due to unprocessed RNA primers of Okazaki fragments, specially for the very short gaps. The data shows a distribution, so that an increase in size due to RNH digestion will enlarge short gaps that would be moved to the large size section. The way to solve this issue is providing the control data of E2 non-treated samples. I think this is important, because the pattern should change dramatically if those gaps reflect hybrids resulting from the burst of transcription after E2 treatment.

We have now provided in the revised manuscript all EM data from the -E2 samples, for each replicate of our EM experiments (Extended Data Fig. 4h-i). As discussed for other data included in the revised Fig. 3, we now have several lines of evidence suggesting that MCF7 cells deprived of estrogen (-E2) experience a peculiar form of replication stress, which is most likely related to their known dependency of estrogen for regular growth. Besides the detectable effect of RNase H1 overexpression discussed above (Fig. 3a), we also observed a mild ZRANB3-dependent fork slowing (Fig. 3b), and accumulation of post-replicative ssDNA gaps which persist for some time behind moving replication forks (Fig. 3g and Extended Data Fig. 4h-i).

Nonetheless, we can now clearly show that *in vitro* RNase H treatment in -E2 samples induces no detectable increase in "pathological" ssDNA gaps (i.e. large and/or distant from forks), while it does induce a reproducible accumulation upon the transcriptional burst (+E2; this aspect is emphasized in the new graph in Fig. 3e). Thus, despite the limitations of -E2 samples as negative controls, our refined analysis of the EM data allows to confidently conclude that an additional category of intermediates (DNA:RNA hybrids embedded in newly replicated duplexes) are specifically associated with the transcriptional burst.

Importantly, this key conclusion of the study is now strongly and independently supported by a new set of data, taking advantage of EdU alkaline comet assays (Fig. 3f-l and Extended Data Fig. 4j-m), showing that various conditions previously associated with DNA:RNA hybrid accumulation and associated fork slowing have defects in nascent strand maturation, and expose persistent discontinuities on their newly replicated duplexes.

6. Fig. 4d-e. The control -IPTG is required. The increase of reversed forks in *drnhc* bacteria is meaningful if not observed in the wt. Otherwise, conclusions are not solid.

While we agree with this reviewer that this may have been an important, additional control, this is unfortunately not technically feasible. As now better explained in the text, differently from the other three conditions, the lack of fork slowing/accumulation in the conflict region in WT - IPTG does not allow to confidently identify replication intermediates from this region, distinguishing them from bulk intermediates from other genomic loci that happen to comigrate in the same area of the gel (see Fig. 4c). In these conditions it is thus not possible to perform reliable EM analyses.

7. Fig. 4d-e. It is unclear from the data the increase observed in the plot of reversed forks in 4e in *drnhc* cells, since the data in 4d shows 11/70 versus 14/70 reversed forks containing fibers. I assume I am missing something. Explain, please.

We have now better specified in the figure legend that Fig. 4d displays intermediates from a single EM experiment (as representative example), but that our conclusions are based on the analysis of three biological replicates and statistical analyses (Fig. 4e). We apologize for this confusion in the submitted manuscript.

8. Fig. 5d-f. Here is where the problem of Okazaki fragments becomes more important. The only way to detect the hybrids in those experiments was by S9.6 labeling. Since the displaced ssDNA seems not to be really detected, the assumption would be that hybrids are very short, in contrast with the data *in vitro* and from HeLa cells (Figs. 1

and 2). This together with the overabundance of gaps in the daughter DNAs makes likely that a good portion of such signals correspond to Okazaki primers rather than newly generated hybrids. Thus, it is essential to show at least the analysis of -IPTG samples to check whether indeed such daughter S9.6 signal overabundance is observed.

We thank the reviewer for this important remark. As discussed above, analysis of the -IPTG samples is not technically possible, but we have addressed this important concern differently. As intrinsic intermediate of the replication process, Okazaki fragments are expected at all replication intermediates, also beyond the conflict region. We thus extended our analysis to replication intermediates extracted from “bulk DNA”, i.e. all other genomic DNA fragments that are digested in fragments of different size than the conflict region, and that were previously shown not to carry detectable amounts of DNA:RNA hybrids (Lang et al., Cell 2017). Importantly, we did observe background levels of S9.6-Gold binding on these intermediates, but this did not significantly differ from linear bulk fragments or from levels of binding observed on the conflict intermediates after extensive RNase H digestion. Accordingly, RNase H digestion of bulk replication intermediates did not reveal any marked accumulation of ssDNA regions, confirming that extended DNA:RNA hybrids are not detectably present. These crucial controls prove that Okazaki fragments are not widely recognized by S9.6-Gold antibody in our experimental conditions and strongly support that post-replicative DNA:RNA hybrids are a specific feature of replication intermediates at transcription-replication conflicts, i.e. a key conclusion of our study.

9. Fig. 5e legend. It refers to Extended Data Fig. 7 when it should be 5.

We thank the reviewer for noticing this, which has been corrected in the revised manuscript.

Discussion

1. Major differences between promoting hybrid formation via estrogen, in which a burst of transcription would cause hybrids that would not form that easily in wt cells, and hybrids detected in rnhC mutants (certainly under transcription conditions established by IPTG) that reveals hybrids formed normally in WT cells that are not processed due to the lack of RNH. Thus, authors are putting together two set of results that may not be related and indeed show differences, so that conclusions may not be unique to all. This needs to be taken into account and discussed properly.

We agree with this remark. We have now added a “disclaimer” sentence in the opening paragraph of the discussion, stressing that – despite the similar molecular phenotypes observed in our study across species and different perturbations of RNA accumulation/processing – other specific perturbations of these mechanisms may lead to drastically different scenarios, in terms of replication interference and intermediates.

2. Authors should discuss possible differences between bacteria and HeLa cells and between hybrids behind the replication fork and the rest. With the issue of Okazaki’s the conclusion that hybrids are formed or accumulated more often is not convincing. Please, keep in mind that hybrid length has been established with the bisulfite mutagenesis by different authors in the Ig genes, yeast or Hela cells. This technique relies on the sequencing of the mutagenized displaced ssDNA, which means they detect R loops. The study in ref. 46 shows a nice correlation between S9.6 maps and bisulfite maps, abounding in the idea that most hybrids seem to be in the form of R loops. This is consistent with in the in vitro and Hela +E2 system is where clearly R loops can be visualized, but contrasts with the conclusions here referring to hybrids as the most prominent event and contributing to the uncertainty about the nature of the assymetric distribution of S9.6 signals around forks, which could be partially explained by Okazaki primers. This needs to be considered and discussed, and the control experiments suggested should help clarify.

Our study does display both canonical R-loops and DNA:RNA hybrids as detectable intermediates under experimental conditions previously linked to R-loop accumulation. As such, our data do not disprove, but rather confirm that both types of intermediates are present in cells and are physio-pathologically relevant, as suggested by multiple studies in this area. However, a key conclusion which we consider strongly supported by the data – also in light of the latest controls we provide regarding Okazaki fragment recognition – is that even in conditions that do lead to accumulation of canonical R-loops, those do not seem *per se* to stall replication forks. Rather,

replication interference is specifically associated with detectable accumulation of DNA:RNA hybrids behind those slowly-moving forks, implying post-replicative mechanisms in the observed replication interference. While refraining from any relative genome-wide quantification of R-loops vs DNA:RNA hybrids – which is not technically possible with our EM method – we are convinced that several lines of evidence (consolidated in revision) support a central role for post-replicative hybrids in TRCs and focused the Discussion on this key conclusion of our study.

3. Authors affirms that transcription is silenced behind the fork. However, this has been defined in eukaryotes in connection to chromatin structure (ref. 48). How would this silencing be achieved in *Bacillus*? Provide a reference or clarify this issue.

To our knowledge this has not been tested thoroughly and systematically in bacteria. We have now specified in the corresponding sentence that this interpretation specifically refers to eukaryotic systems.

4. This is an impressive amount of great work, but it would be nice in the future to study the co-directional system to probe part of the models. Since it is well known that the impact of head-on is much higher than the codirectional, it is not easy to understand why the codirectional hybrids, accumulated at the future leading strand will not suppose a block to the progression of the fork, while head-on will do if most of hybrids are found behind the fork, from which they are supposed to delay replication. The RNA polymerase is the first physical obstacle that the fork encounters in head-on. Could it be that the pause of the fork itself may contribute to the formation of the short hybrids behind the fork as a consequence of unprocessed Okazaki primers?. It would be nice if authors can add some light with their Discussion on this fact.

We do agree with the reviewer that performing similar experiments in the co-directional system could provide important clues. However, we note that, so far, this would only be possible in the *B. subtilis* system, which may represent a key limitation to effectively address some of the interesting questions raised by the referee. We envision several possibilities on how our findings may “translate” in the co-directional scenario, but feel that contemplating these different scenarios as part of our Discussion (unavoidably already complex) would dilute the key messages and possibly confuse most readers. Moreover, proper discussion of different scenarios would require extensive graphical representation for clarity, which seems most appropriate for a review article than for the Discussion of our work. In light of this, we have simply added a sentence in the Discussion, highlighting the importance to perform similar analyses on co-directional TRCs in the future.

5. Along with their line of thinking, authors should include in their Discussion the recent paper showing indeed that hybrids may have different origins and processed by different factors PMID: 34294712

We have now included a citation of this work in the relevant sentence of the Discussion.

6. It has recently been published (PMID: 33657379) that topological stress generated at head-on collisions causes R loops ahead of the fork, events that are controlled by type II topoisomerases in *Bacillus*. This new study, however, shows in the same bacteria sp, that hybrids seems to be poorly seen ahead of the fork. Any hints? A discussion would be nice, because it is not clear at all how a hybrid behind the fork should delay fork movement, as its connection with reversed forks is not too clear, since most S9.6 signals (Fig. 5 and Extended Fig.5 shows most S9.6 signal not associated with fork reversal)

Genetic and biochemical analyses included in the report mentioned by the reviewer (which was already discussed in our submitted manuscript) actually point towards a wide accumulation of torsional stress – and footprint of topoisomerase action – within and around the head-on conflict region, rather than showing specific accumulation ahead of replication forks. As mentioned in the Discussion, these data in fact offer further support to a redistribution of topological constraints behind the slowly moving replication forks, rather than pointing to specific accumulation of torsional stress between hybrid and approaching fork. This is entirely consistent with fork slowdown, fork reversal and post-replicative hybrids being detectable across the entire conflict region (Figs. 4 and 5), rather than accumulating at a specific location within the fragment of interest. However, we acknowledge that the mechanisms mediating the slowdown after initial bypass are currently unclear and we do present alternative possibilities in the Discussion.

7. ref 27 should be cited when referring to model in Extended Fig. 7a 1. In that reference it was not proposed that the fork is able to pass the hybrids, as the authors indicate in Intro and Discussion (please correct), but that indeed hybrids are formed behind the forks in the lagging strand with an RNA molecule produced in the leading strand and if so, such putative hybrids behind the fork not affecting fork movement. This needs to be properly referred.

We apologize for this inadvertent confusion. We have now appropriately referred to this paper in the relevant sentence of the Discussion. The reference to it in the Introduction does not actually refer to hybrid bypass and seems accurate as it stands.

8. One interesting proposal in this study is that hybrids seen ahead of the fork may be the product of a previous R loop in which the displaced ssDNA is removed. Do authors think that NER endonucleases would do this as has been shown (ref. 18). Has anybody studied TRCs in UvrC mutants in Bacillus? Wpuld result fit with that notion? This is an interesting thought, on which however we can provide no direct evidence. Moreover, we are not aware of any study on UvrC in *B. subtilis* in the context of R-loops. Testing multiple potential R-loop processing enzymes for their contribution to replication bypass seems like an exciting perspective for future studies, although it may require less time-consuming methods than EM analyses described here. In the absence of any direct data or specific suggestion from the literature, we refrained from any comment on this in the Discussion.

In summary, excellent work. I hope my comments contribute to make it stronger.

Reviewer #3:

Remarks to the Author:

This is a nicely written manuscript describing a novel electron microscopy (EM)-based approach to directly visualize R-loops formed during head-on transcription replication conflicts. The authors start by showing that R-loops can be visualized in vitro by EM and that these structures are characterized by a ssDNA strand, which is displaced from the RNA:DNA duplex. Conversely, RNA:DNA hybrid filaments cannot be distinguished from the surrounding DNA by EM. However, they show that RNA:DNA hybrids can be detected by EM using an S9.6-gold labeling approach. The same EM-based approach was then successfully applied to visualize R-loops in a mammalian breast cancer cell line (MCF7), where estrogen stimulation causes a transcriptional burst that leads to R-loop accumulation. From this analysis, the authors conclude that larger R-loops, longer than 300bp, are more frequently induced by the transcriptional burst compared to smaller R-loops, although the molecular basis for this difference remain unclear. Using this new approach, the authors estimate the total R-loop burden on the genome caused by the same transcriptional burst. Next, they investigate the effect of aberrant R-loop accumulation on DNA replication. In agreement with previous findings, they observe that the transcriptional burst perturbs replication fork progression and suggest that this fork slowing is associated with reversed fork formation because the fork slowing phenotype is rescued by PARP inhibition. By inspecting the structure of the replication intermediates by EM, they also observe increased accumulation of ssDNA gaps behind the replication forks upon treatment with RNase H, suggesting that the transcriptional burst leads to hybrid accumulation behind the forks. Finally, they expand their analysis to study how RNA:DNA hybrids interfere with DNA replication in a bacterial model system. From this analysis, they confirm that replication forks frequently reverse when facing transcription replication conflicts and that RNA:DNA hybrids accumulate behind the replication forks, suggesting that these structures are efficiently bypassed by the replisome. Overall, this an important study that provides the first direct visualization of R-loop structures in cells and new insights

into the mechanisms by which replication forks deal with these intermediates. I think that these findings will spark the interest of many groups studying transcription replication collisions and will open new avenues of research in the field.

We thank this reviewer for the positive evaluation of our work, its relevance, and its potential impact on the field.

However, the authors need to include additional experiments and missing controls to strengthen their conclusions and increase the impact of their findings.

Major criticisms:

1. The extended data of Figure 1E shows almost 50% nonspecific S9.6 binding when using 50% of the competitor the plasmid that lacks R-loops. This result undermines the interpretation of the following results obtained with the S9.6 antibody, particularly the results of Figure 5d where the authors map the location of RNA:DNA hybrids on replication forks using the S9.6 label.

We highlighted and discussed the limitations of the use and specificity of the S9.6 antibody, which are also by now well known in the field. However, we note that key conclusions of our study stem from the use of S9.6Gold under controlled conditions of specificity (Fig. 5). Please note that, spiking our samples with an excess of epitope-containing plasmid (pFC53), we demonstrably retained S9.6-Gold specificity while analyzing its binding to replication intermediates; indeed binding to pFC53 within the mix containing replication intermediates (Fig. 5c) is perfectly in line with the binding levels observed on the pure plasmid in Fig. 1d and is still largely RNase H-sensitive (Fig. 5c). Based on these data, we can reasonably assume that S9.6-Gold binding is still highly specific for DNA:RNA hybrids in the experimental conditions used to reveal postreplicative hybrids on replication intermediates of the conflict region (Fig. 5d).

2. Figure 2. The authors should discuss the possibility that longer R-loops are more reproducibly increased compared to shorter R-loops simply because they might be more stable and more likely to persist during the DNA extraction process compared to shorter R-loops.

We cannot directly test differential stability of long vs short R-loops. However, we note that the vast majority of the observed R-loops on genomic DNA is smaller than 200bp, arguing against a stability issue for smaller R-loops. We are thus convinced that the relative increase in long R-loops upon E2 addition most likely reflects a true increase of these structures in this biological condition. We have however mentioned in the corresponding section of the results that increased stability (or detectability) of these structures may contribute to their observed accumulation upon E2 addition.

3. The two replicates of the experiments showing the frequency of small R-loops (Extended Data Figure 3e) show very different results. In particular, the first repeat does not show any significant increase in R-loops after estrogen stimulation. These experiments should be repeated in triplicate to confirm that the results are reproducible.

We do acknowledge (here, and directly in the main text of the manuscript) that the two biological replicates yielded slightly different results concerning short R-loops, while confirming the result for longer ones. In light of the fact that these experiments are extremely time consuming, and that several additional experiments had to be prioritized for this revision, we could not perform a third replicate of this experiment. We note, however, that this was unlikely to change the conclusion that effects on smaller R-loops are in the best case variable, across multiple biological replicates. As the focus of this manuscript was on the role of DNA:RNA hybrids in transcription-replication interference, we consider this as a minor shortcoming of our dataset, which does not affect any of its key conclusions.

4. Figure 3. The authors should test whether the fork slowing phenotype associated with estrogen induction is rescued by addition of RNase H to provide further support to their model that this phenotype is linked to accumulation of RNA:DNA hybrids.

We have now performed the experiment suggested by this reviewer, transiently overexpressing RNase H1 in MCF7 cells. Importantly, even moderate expression of RNase H1 in replicating cells (Extended Data Fig. 4a) leads to a marked rescue of fork speed in presence of E2, clearly linking DNA:RNA hybrid accumulation with delayed fork progression, and thereby consolidating a key conclusion of our manuscript.

5. Figure 3. The authors' model that RNase H treatment leads to accumulation of ssDNA gaps behind the replication forks should be strengthened by additional DNA fiber assays in the presence of the S1 nuclease, which would cause tract shortening if ssDNA gaps are present on the thymidine labeled tracts.

We thank this reviewer for this interesting suggestion, but note that such experiment would have required a careful titration of RNase H and S1 treatments *in vitro*, in order to specifically reveal hybrid-related discontinuities on replicated DNA. We have however set out to address this important point with a different method. We optimized a recently published method to reveal discontinuities on nascent DNA and developed an EdU-alkaline comet assay, which is sensitive enough to reveal physiological discontinuities during DNA synthesis (i.e. during lagging strand synthesis) and their rapid ligation behind unperturbed replication forks (Extended Data Fig. 3j). Using this method in various experimental conditions previously associated with DNA:RNA hybrid accumulation and associated fork slowing, we now show that hybrid accumulation is consistently linked to delays in nascent strand maturation, and expose persistent discontinuities on their newly replicated duplexes (Fig. 3g-i). We consider this an essential addition to our revised manuscript, which provides significant support to one of the key conclusions of our study and which has been highlighted in our revised Discussion.

6. Figure 3a. PARP has several cellular functions, in addition to the previously reported role in reversed fork stabilization discussed in the manuscript. The authors should repeat the DNA fiber experiments of Figure 3a by depleting the SMARCAL1 translocase to prevent formation of reversed forks and provide further support to their model that fork slowing is indeed associated with reversed fork accumulation. The same experiments could be extended to other translocases involved in fork reversal such as ZRANB3 or HLTF. Moreover, the authors should repeat the same assays by depleting the RECQ1 helicase, which should prevent reversed fork stabilization. We have addressed this concern by transient downregulation of ZRANB3, while depletion of SMARCAL1 proved not compatible with unperturbed cell cycle progression of MCF7 cells (likely supporting the view that these cells experience some form of replication stress even in the absence of E2). Importantly, ZRANB3 downregulation largely rescued the slowdown of fork progression observed upon E2 addition, consolidating the link between fork slowing and fork reversal in these conditions. In light of the (agreeable) criticism of this reviewer to the PARPi experiments, those have been moved to Extended Data Fig. 4, and are used in the revised manuscript to consolidate the key conclusion stemming from data in Fig. 3 (dependency of fork slowing upon DNA:RNA hybrids and fork reversal).

7. Figure 3a. The authors should measure the length of IdU tracts and confirm that the transcriptional burst leads to the same decrease in tract length observed when measuring the CldU tracts.

For unknown reasons, MCF7 cells have issues incorporating IdU at similar rates as CldU, and higher IdU concentrations of this analog are toxic to these cells. Thus, although we observed similar trends while measuring IdU (instead of CldU) tracks in our experiments (Fig. 1 for reviewers only), we overall consider the length of these tracks as less reliable than CldU tracks for quantitative measurements. Hence, while double labelling was still useful to identify ongoing forks, we preferred to limit fork speed analysis to CldU tracks.

Fig. 1 for reviewers only. DNA fiber assay of MCF7 +/- E2, combined with 96h of siZRANB3 transfection prior to E2 treatment. Quantification of IdU tract lengths [μm]; at least 100 individual molecules quantified per condition. Median fiber length indicated in red. Despite comparable labeling time (see Fig. 3b) and reproducible trends for CldU and IdU tracks, IdU track length is ca. 50% shorter than CldU tracks (compare to Fig. 3b), at nucleotide concentrations tolerated by the cells.

8. Figure 5d shows S9.6 binding to the parental strand and one of the two daughter strands of different replication intermediates. An additional figure showing the same results for the complementary daughter strand would be informative to help the reader understanding whether S9.6 binding can simultaneously occur on both strands. While we in principle agree that this may be valuable information, intrinsic limitations of our EM analysis make this representation rather confusing and uninformative. We are unfortunately unable to distinguish leading and lagging strands on replication intermediates, making the assignment (color coding) of the daughter stands arbitrary, non-informative and possibly even misleading. We provide here (Fig. 2 for reviewers only) an example of such analysis (dmhc + IPTG); although roughly half of the replication intermediates are bound by S9.6-gold, only rarely we observe multiple binding, impairing any sensible analysis of statistically significant trends. Hence, we prefer not to show this type of representation, which affects readability of our maps, without providing significant additional information to the readers

Fig. 2 for reviewers only. S9.6-Gold binding position within the replicating conflict of *B. subtilis*. Green: daughter strand. Grey/black: parental strand. Pink: regressed arm. Red/blue: S9.6-Gold labels on Daughter strands 1 and 2, arbitrarily numbered.

9. For the experiments in *Bacillus subtilis*, the authors state that “the delta-rnhc mutant strain lacks endogenous RNase H activity”. However, the authors should clarify that this strain only lacks RNase H III activity, and still retains some residual RNase H activity mediated by RNase HII.

We thank the reviewer for this remark. We have corrected those statements for accuracy.

10. Some of the figures are missing essential controls. In particular, Figure 3d should include the untreated control (-E2). The same applies to Figure 4d that lacks the (WT-IPTG) control.

We have now provided in the revised manuscript all EM data from the -E2 samples, for each replicate of our EM experiments (Extended Data Fig. 4h-i). As discussed in several parts of the revised manuscript, we now have several lines of evidence suggesting that MCF7 cells deprived of estrogen (-E2) experience a peculiar form of replication stress, which is most likely related to their known dependency of estrogen for regular growth. Even in the absence of E2, we observed detectable effects of RNase H1 overexpression on fork speed (Fig. 3a), a mild ZRANB3-dependent fork slowing (Fig. 3b), and accumulation of post-replicative ssDNA gaps which persist for some time behind moving replication forks (Fig. 3g and Extended Data Fig. 4hi). Nonetheless, we can now clearly show that in vitro RNase H treatment in -E2 samples induces no detectable increase in “pathological” ssDNA gaps (i.e. large and/or distant from forks), while it does induce a reproducible accumulation upon the transcriptional burst (+E2; this aspect is emphasized in the new graph in Fig. 3e). Thus, despite the limitations of -E2 samples as negative controls, our refined analysis of the EM data allows to confidently conclude that an additional category of intermediates (DNA:RNA hybrids embedded in newly replicated duplexes) are specifically associated with the transcriptional burst.

Regarding the WT-IPTG control in Figure 4, while we agree with this reviewer that this may have been an important additional control, this is unfortunately not technically feasible. As now better explained in the text, differently from the other three conditions, the lack of fork slowing/accumulation in the conflict region in WT - IPTG does not allow to confidently identify replication intermediates from this region, distinguishing them from bulk intermediates from other genomic loci that happen to comigrate in the same area of the gel (see Fig. 4c). In these conditions it is thus not possible to perform reliable EM analyses.

Minor comments:

1. Figure 2d is not referenced in the text.

This has been corrected in the revised manuscript

Decision Letter, first revision:

Message: Our ref: NSMB-A45242A-Z

6th Dec 2022

Dear Dr. Lopes,

Thank you for submitting your revised manuscript "Direct visualization of transcription-replication conflicts reveals post-replicative DNA:RNA hybrids" (NSMB-A45242A-Z). It has now been seen by the original referees and their comments are below. The reviewers find that the paper has improved in revision, and therefore we'll be happy in principle to publish it in Nature Structural & Molecular Biology, pending minor revisions to satisfy the referees' final requests and to comply with our editorial and formatting guidelines.

To facilitate our work at this stage, we would appreciate if you could send us the main text as a word file. Please make sure to copy the NSMB account (cc'ed above).

Sincerely,

Carolina

Carolina Perdigoto, PhD
Chief Editor
Nature Structural & Molecular Biology
orcid.org/0000-0002-5783-7106

Reviewer #2 (Remarks to the Author):

Authors have largely improved the manuscript adding new experiments requested and clarifying conclusions. Technical limitations have been properly explained by authors. The manuscript is a nice piece of work that merits publication and will be of relevance for the field.

Reviewer #3 (Remarks to the Author):

The authors properly addressed most of the previous comments of this reviewer. The revised version of the manuscript is significantly improved and suitable for publication.

Minor criticism:

1. In response to point #7 of this reviewer, the authors stated that they experienced issues incorporating IdU at similar rates as CldU in the MCF7 cells. The authors should report this issue in the "Methods" section of the manuscript, as this is an important limitation of the DNA fiber assay that needs to be disclosed to the readers.

Author Rebuttal, first revision:

Reviewer #2:

Remarks to the Author:

Authors have largely improved the manuscript adding new experiments requested and clarifying conclusions. Technical limitations have been properly explained by authors. The manuscript is a nice piece of work that merits publication and will be of relevance for the field.

Reviewer #3:

Remarks to the Author:

The authors properly addressed most of the previous comments of this reviewer. The revised version of the manuscript is significantly improved and suitable for publication.

Minor criticism:

1. In response to point #7 of this reviewer, the authors stated that they experienced issues incorporating IdU at similar rates as CldU in the MCF7 cells. The authors should report this issue in the "Methods" section of the manuscript, as this is an important limitation of the DNA fiber assay that needs to be disclosed to the readers.

The following statement has been added to the corresponding method section: "Of note, we noticed that MCF7 cells experience issues incorporating IdU and therefore considered the CldU tract length as the more reliable readout for these fiber experiments. "

Final Decision Letter:**Message** 23rd Jan 2023

:

Dear Prof. Lopes,

We are now happy to accept your revised paper "Direct visualization of transcription-replication conflicts reveals post-replicative DNA:RNA hybrids" for publication as a Article in Nature Structural & Molecular Biology.

As soon as your article is published, you can generate your shareable link by entering the DOI of your article here: http://authors.springernature.com/share. Corresponding authors will also receive an automated email with the shareable link

Your paper will be published online soon after we receive proof corrections and will appear in print in the next available issue. You can find out your date of online publication by contacting the production team shortly after sending your proof corrections. Content is published online weekly on Mondays and Thursdays, and the embargo is set at 16:00 London time (GMT)/11:00 am US Eastern time (EST) on the day of publication. Now is the

time to inform your Public Relations or Press Office about your paper, as they might be interested in promoting its publication. This will allow them time to prepare an accurate and satisfactory press release. Include your manuscript tracking number (NSMB-A45242B) and our journal name, which they will need when they contact our press office.

About one week before your paper is published online, we shall be distributing a press release to news organizations worldwide, which may very well include details of your work. We are happy for your institution or funding agency to prepare its own press release, but it must mention the embargo date and Nature Structural & Molecular Biology. If you or your Press Office have any enquiries in the meantime, please contact press@nature.com.

Please note that *Nature Structural & Molecular Biology* is a Transformative Journal (TJ). Authors may publish their research with us through the traditional subscription access route or make their paper immediately open access through payment of an article-processing charge (APC). Authors will not be required to make a final decision about access to their article until it has been accepted. [Find out more about Transformative Journals](https://www.springernature.com/gp/open-research/transformative-journals)

Authors may need to take specific actions to achieve [compliance with funder and institutional open access mandates](https://www.springernature.com/gp/open-research/funding/policy-compliance-faqs). If your research is supported by a funder that requires immediate open access (e.g. according to [Plan S principles](https://www.springernature.com/gp/open-research/plan-s-compliance)) then you should select the gold OA route, and we will direct you to the compliant route where possible. For authors selecting the subscription publication route, the journal's standard licensing terms will need to be accepted, including [self-archiving policies](https://www.springernature.com/gp/open-research/policies/journal-policies). Those licensing terms will supersede any other terms

that the author or any third party may assert apply to any version of the manuscript.

Sincerely,
Dimitris

Dimitris Typas, PhD
Associate Editor
Nature Structural & Molecular Biology
ORCID: 0000-0002-8737-1319
